# Efficient Inference and Exploration for Reinforcement Learning

## Abstract

Despite an ever growing literature on reinforcement learning algorithms and applications, much less is known about their statistical inference. In this paper, we investigate the large-sample behaviors of the Q-value estimates with closed-form characterizations of the asymptotic variances. This allows us to efficiently construct confidence regions for Q-value and optimal value functions, and to develop policies to minimize their estimation errors. This also leads to a policy exploration strategy that relies on estimating the relative discrepancies among the Q estimates. Numerical experiments show superior performances of our exploration strategy than other benchmark approaches.

## 1 Introduction

We consider the classical reinforcement learning (RL) problem where the agent interacts with a random environment and aims to maximize the accumulated discounted reward over time. The environment is formulated as a Markov decision process (MDP) and the agent is uncertain about the true dynamics to start with. As the agent interacts with the environment, data about the system dynamics are collected and the agent becomes increasingly confident about her decision. With finite data, however, the potential reward from each decision is estimated with errors and the agent may be led to a suboptimal decision. Our focus in this paper is on statistically efficient methodologies to quantify these errors and uncertainties, and to demonstrate their use in obtaining better policies.

More precisely, we investigate the large-sample behaviors of estimated Q-value, optimal value function, and their associated policies. Our results are in the form of asymptotic convergence to an explicitly identified and computable Gaussian (or other) distribution, as the collected data sizes increase. The motivation of our investigation is three-fold. First, these precise asymptotic statements allow us to construct accurate confidence regions for quantities related to the optimal policy, and, like classical statistical inference, they can assess the reliability of the current estimates with respect to the data noises. Second, our results complement some finite-sample error bounds developed in the literature (Kearns & Singh, 1998; Kakade, 2003; Munos & Szepesvári, 2008), by supplementing a closed-form asymptotic variance that often shows up in the first-order terms in these bounds.

Our third and most important motivation is to design good exploration policies by directly using our tight error estimates. Motivated by recent autonomous-driving and other applications (e.g., Kalashnikov et al. (2018)), we consider the pure exploration setting where an agent is first assigned an initial period to collect as much experience as possible, and then, with the optimal policy trained offline, starts deployment to gain reward. We propose an efficient strategy to explore by optimizing the worst-case estimated relative discrepancy among the Q-values (ratio of mean squared difference to variance), which provides a proxy for the probability of selecting the best policy. Similar criteria have appeared in the so-called optimal computing budget allocation (OCBA)

procedure in simulation-based optimization (Chen & Lee, 2011) (a problem closely related to best-arm identification (Audibert & Bubeck, 2010) in online learning). In this approach, one divides computation (or observation) budget into stages in which one sequentially updates mean and variance estimates, and optimizes next-stage budget allocations according to the worst-case relative discrepancy criterion. Our proposed procedure, which we term Q-OCBA, follows this idea with a crucial use of our Q-value estimates and randomized policies to achieve the optimal allocation. We demonstrate how this idea consistently outperforms other benchmark exploration policies, both in terms of the probability in selecting the best policy and generating the tightest confidence bounds for value estimates at the end of the exploration period.

Regarding the problem of constructing tight error estimates in RL, the closest work to ours is Mannor et al. (2004; 2007), which studies the bias and variance in value function estimates with a fixed policy. Our technique resolves a main technical challenge in Mannor et al. (2004; 2007), which allows us to substantially generalize their variance results to Q-values, optimal value functions and asymptotic distributional statements. The derivation in Mannor et al. (2004; 2007) hinges on an expansion of the value function in terms of the perturbation of the transition matrix, which (as pointed out by the authors) is not easily extendable from a fixed-policy to the optimal value function. In contrast, our results utilize an implicit function theorem applied to the Bellman equation that can be verified to be sufficiently smooth. This idea turns out to allow us to obtain gradients for Q-values, translate to the optimal value function, and furthermore generalize to similar results for constrained MDP and approximate value iterations. We also relate our work to the line of studies on dynamic treatment regimes (DTR) (Laber et al., 2014) applied commonly in medical decision-making, which focuses on the statistical properties of polices on finite horizon (such as two-period). Our infinite-horizon results on the optimal value and Q-value distinguishes our developments from the DTR literature. Moreover, our result on the non-unique policy case can be demonstrated to correspond to the "non-regularity" concept in DTR, where the true parameters are very close to the decision "boundaries" that switch the optimal policy (motivated by situations of small treatment effects), thus making the obtained policy highly sensitive to estimation noises.

In the rest of this paper, we first describe our MDP setup and notations (Section 2). Then we present our results on large-sample behaviors (Section 3), demonstrate their use in exploration strategies (Section 4), and finally substantiate our findings with experimental results (Section 5). In the Appendix, we first present generalizations of our theoretical results to constrained MDP (A.1) and problems using approximate value iteration (A.2). Then we include more numerical experiments (B), followed by all the proofs (C).

## 2 PROBLEM SETUP

Consider an infinite horizon discounted reward MDP, $\mathcal{M} = (\mathcal{S}, \mathcal{A}, R, P, \gamma, \rho)$, where $\mathcal{S}$ is the state space, $\mathcal{A}$ is the action space, $R(s, a)$ denotes the random reward when the agent is in state $s \in \mathcal{S}$ and selects action $a \in \mathcal{A}$, $P(s'|s, a)$ is the probability of transitioning to state $s'$ in the next epoch given current state $s$ and taken action $a$, $\gamma$ is the discount factor, and $\rho$ is the initial state distribution. The distribution of the reward $R$ and the transition probability $P$ are unknown to the agent. We assume both $\mathcal{S}$ and $\mathcal{A}$ are finite sets. Without loss of generality, we denote $\mathcal{S} = \{1, 2, \ldots, m_s\}$ and $\mathcal{A} = \{1, 2, \ldots, m_a\}$. Finally, we make the following stochasticity assumption:

**Assumption 1.** *$R(s, a)$ has finite mean $\mu_R(s, a)$ and finite variance $\sigma_R^2(s, a)$ $\forall$ $s \in \mathcal{S}, a \in \mathcal{A}$. For any given $s \in \mathcal{S}$ and $a \in \mathcal{A}$, $R(s, a)$ and $S' \sim P(\cdot|s, a)$ are all independent random variables.*

A policy $\pi$ is a mapping from each state $s \in \mathcal{S}$ to a probability measure over actions $a \in \mathcal{A}$. Specifically, we write $\pi(a|s)$ as the probability of taking action $a$ when the agent is in state $s$ and $\pi(\cdot|s)$ as the $m_a$-dimensional vector of action probabilities at state $s$. For convenience, we sometimes write $\pi(s)$ as the

realized action given the current state is $s$. The value function associated with a policy $\pi$ is defined as $V^\pi(s) = \mathbb{E}[\sum_{t=0}^\infty \gamma^t R(s_t, \pi(s_t))|s_0 = s]$ with $s_{t+1} \sim P(.|s_t, \pi(s_t))$. The expected value function, under the initial distribution $\rho$, is denoted by $\chi^\pi = \sum_s \rho(s) V^\pi(s)$. A policy $\pi^*$ is said to be optimal if $V^{\pi^*}(s) = \max_\pi V^\pi(s)$ for all $s \in \mathcal{S}$. For convenience, we denote $V^* = V^{\pi^*}$ and $\chi^* = \sum_s \rho(s) V^*(s)$. The Q-value, denoted by $Q(s, a)$, is defined as $Q(s, a) = \mu_R(s, a) + \gamma \mathbb{E}[V^*(S')|s, a]$. Correspondingly, $V^*(s) = \max_a Q(s, a)$ and the Bellman equation for $Q$ takes the form

$$Q(s, a) = \mu_R(s, a) + \gamma \mathbb{E}\left[\max_{a'} Q(s', a')|s, a\right], \tag{1}$$

for any $(s, a) \in \mathcal{S} \times \mathcal{A}$. Denoting the Bellman operator as $\mathcal{T}_{\mu_R, P}(\cdot)$, $Q$ is a fixed point associated with $\mathcal{T}_{\mu_R, P}$, i.e. $Q = \mathcal{T}_{\mu_R, P}(Q)$.

For the most part of this paper we make the following assumption about $Q$:

**Assumption 2.** *For any state $s \in \mathcal{S}$, $\arg\max_{a \in \mathcal{A}} Q(s, a)$ is unique.*

Under Assumption 2, the optimal policy $\pi^*$ is unique and deterministic. Let $a^*(s) = \arg\max_{a \in \mathcal{A}} Q(s, a)$. Then $\pi^*(a|s) = \mathbb{1}(a = a^*(s))$, where $\mathbb{1}(\cdot)$ denotes the indicator function.

We next introduce some statistical quantities arising from data. Suppose we have $n$ observations (whose collection mechanism will be made precise later), which we denote as $\{(s_t, a_t, r_t(s_t, a_t), s'_t(s_t, a_t)) : 1 \le t \le n\}$, where $r_t(s_t, a_t)$ is the realized reward at time $t$ and $s'_t(s_t, a_t) = s_{t+1}$. We define the sample mean $\hat\mu_{R,n}$ and the sample variance $\hat\sigma^2_{R,n}$ of the reward as

$$\hat\mu_{R,n}(s = i, a = j) = \frac{\sum_{1 \le t \le n} r_t(s_t, a_t)\mathbb{1}(s_t = i, a_t = j)}{\sum_{1 \le t \le n} \mathbb{1}(s_t = i, a_t = j)}, \tag{2}$$

$$\hat\sigma^2_{R,n}(s = i, a = j) = \frac{\sum_{1 \le t \le n} r_t(s_t, a_t)^2 \mathbb{1}(s_t = i, a_t = j)}{\sum_{1 \le t \le n} \mathbb{1}(s_t = i, a_t = j)} - \hat\mu_{R,n}(i, j)^2. \tag{3}$$

Similarly, we define the empirical transition matrix $\hat{P}_n$ as

$$\hat{P}_n(s' = k|s = i, a = j) = \frac{\sum_{1 \le t \le n} \mathbb{1}(s_t = i, a_t = j, s'_t(s_t, a_t) = k)}{\sum_{1 \le t \le n} \mathbb{1}(s_t = i, a_t = j)} \tag{4}$$

and its $m_s \times m_s$ sampling covariance matrix $\Sigma_{P_{s,a}}$ (with one sample point of $\mathbb{1}(s_t = s, a_t = a)$) as

$$\Sigma_{P_{s,a}}(k_1, k_2) = \begin{cases} P(k_1|s, a)(1 - P(k_1|s, a)) & k_1 = k_2 \\ -P(k_1|s, a)P(k_2|s, a) & k_1 \neq k_2. \end{cases}, \text{ for } 1 \le k_1 \le m_s, 1 \le k_2 \le m_s.$$

With the data, we construct our estimate of $Q$, called $\hat{Q}_n$, which is the empirical fixed point of $\mathcal{T}_{\hat\mu_{R,n}, \hat{P}_n}$, i.e. $\hat{Q}_n = \mathcal{T}_{\hat\mu_{R,n}, \hat{P}_n}(\hat{Q}_n)$. Correspondingly, we also write $\hat{V}_n^*(s) = \max_{a \in \mathcal{A}} \hat{Q}_n(s, a)$ and $\hat\chi_n^* = \sum_{s \in \mathcal{S}} \rho(s) \hat{V}_n^*(s)$.

We shall focus on the empirical errors due to noises of the collected data, and assume the MDP or Q-value evaluation can be done off-line so that the fixed point equation for $\hat{Q}_n$ can be solved exactly. .

## 3 Quantifying Asymptotic Estimation Errors

We present an array of results regarding the asymptotic behaviors of $\hat{Q}_n$ and $\hat{V}_n^*$. To prepare, we first make an assumption on our exploration policy $\pi$ to gather data. Define the extended transition probability $\tilde{P}^\pi$ as $\tilde{P}^\pi(s', a'|s, a) = P(s'|s, a)\pi(a'|s')$. We make the assumption:

**Assumption 3.** *The Markov chain with transition probability $\tilde{P}^{\pi}$ is positive recurrent.*

Under Assumption 3, $\tilde{P}^{\pi}$ has a unique stationary distribution, denoted $w$, equal to the long run frequency in visiting each state-action pair, i.e. $w(s,a) = \lim_{n\to\infty} \frac{1}{n} \sum_{1 \leq t \leq n} \mathbb{1}(s_t = i, a_t = j)$, where all $w(s,a)$'s are positive. Note that Assumption 3 is satisfied if for any two states $s, s'$, there exists a sequence of actions such that $s'$ is attainable from $s$ under $P$, and, moreover, if $\pi$ is sufficiently mixed, e.g., $\pi$ satisfies $\pi(a'|s') > 0$ for all $s', a'$.

Our results in the sequel uses the following further notations. We denote "$\Rightarrow$" as "convergence in distribution", and $\mathcal{N}(\mu, \Sigma)$ as a multivariate Gaussian distribution with mean vector $\mu$ and covariance matrix $\Sigma$. We write $I$ as the identity matrix, and $e_i$ as the $i$-th unit vector. The dimension of $\mathcal{N}(\mu, \Sigma)$, $I$ and $e_i$ should be clear from the context. When not specified, all the vectors are column vectors. Let $N = m_s m_a$. In our algebraic derivations, we need to re-arrange $\mu_R$, $Q$ and $w$ as $N$-dimensional vectors. We thus define the following indexing rule: $(s = i, a = j)$ is re-indexed as $(i-1)m_a + j$, e.g. $\mu_R(i,j) = \mu_R((i-1)m_a + j)$. We also need to re-arrange $\tilde{P}^{\pi}$ as an $N \times N$ matrix following the same indexing rule, i.e. $\tilde{P}^{\pi}(i', j'|i, j) = \tilde{P}^{\pi}((i-1)m_a + j, (i'-1)m_a + j')$.

## 3.1 Limit Theorems under Sufficient Exploration

We first establish the asymptotic normality of $\hat{Q}_n$ under exploration policy $\pi$:

**Theorem 1.** *Under Assumptions 1 and 2, if the data is collected according to $\pi$ satisfying Assumption 3, then $\hat{Q}_n$ is a strongly consistent estimator of $Q$, i.e. $\hat{Q}_n \to Q$ almost surely as $n \to \infty$. Moreover,*

$$\sqrt{n}(\hat{Q}_n - Q) \Rightarrow \mathcal{N}(0, \Sigma) \quad as \quad n \to \infty,$$

*where*

$$\Sigma = (I - \gamma\tilde{P}^{\pi^*})^{-1}W^{-1}(D_R + D_Q)((I - \gamma\tilde{P}^{\pi^*})^{-1})^T, \tag{5}$$

*$W$, $D_R$ and $D_Q$ are $N \times N$ diagonal matrices with*

$$W((i-1)m_a + j, (i-1)m_a + j) = w(i,j), \quad D_R((i-1)m_a + j, (i-1)m_a + j) = \sigma_R^2(i,j)$$
$$and \quad D_Q((i-1)m_a + j, (i-1)m_a + j) = (V^*)^T \Sigma_{P_{i,j}} V^* \ respectively .$$

In addition to the asymptotic Gaussian behavior, a key element of Theorem 1 is the explicit form of the asymptotic variance $\Sigma$. This is derived from the delta method (Serfling, 2009) and, intuitively, is the product of the sensitivities (i.e., gradient) of $Q$ with respect to its parameters and the variances of the parameter estimates. Here the parameters are $\mu_R$ and $P$, with corresponding gradients $(I - \gamma\tilde{P}^{\pi^*})^{-1}$ and $(I - \gamma\tilde{P}^{\pi^*})^{-1}V^*$. The variances of these parameter estimates (i.e., (2) and (4)) involve $\sigma_R^2(i,j)$ and $\Sigma_{P_{i,j}}$, and the sample size allocated to estimate each parameter, which is proportional to $w(i,j)$.

Using the relations that $V_n^*(s) = \max_{a \in \mathcal{A}} Q(s,a)$ and $\hat{V}_n^*(s) = \max_{a \in \mathcal{A}} \hat{Q}_n(s,a)$, we can leverage Theorem 1 to further establish the asymptotic normality of $\hat{V}_n^*$ and $\hat{\chi}_n^*$:

**Corollary 1.** *Under Assumptions 1, 2 and 3,*

$$\sqrt{n}(\hat{V}_n^* - V^*) \Rightarrow \mathcal{N}(0, \Sigma_V) \quad and \quad \sqrt{n}(\hat{\chi}_n^* - \chi^*) \Rightarrow \mathcal{N}(0, \sigma_\chi^2) \quad as \quad n \to \infty$$

*where*

$$\Sigma_V = (I - \gamma P^{\pi^*})^{-1}(W^{\pi^*})^{-1}[D_R^{\pi^*} + D_V^{\pi^*}]((I - \gamma P^{\pi^*})^{-1})^T,$$

*$\sigma_\chi^2 = \rho^T \Sigma_V \rho$, $P^{\pi^*}$ is an $m_s \times m_s$ transition matrix with $P^{\pi^*}(i,j) = P(j|s = i, a = a^*(s))$, $W^{\pi^*}$, $D_R^{\pi^*}$ and $D_V^{\pi^*}$ are $m_s \times m_s$ diagonal matrices with $W^{\pi^*}(i,i) = w(i, a^*(i))$, $D_R^{\pi^*}(i,i) = \sigma_R^2(i, a^*(i))$ and $D_V^{\pi^*}(i,i) = (V^*)^T \Sigma_{P_{i,a^*(i)}} V^*$ respectively.*

In the Appendix we also prove, using the same technique as above, a result on the large-sample behavior of the value function for a fixed policy (Corollary 2), which essentially recovers Corollary 4.1 in Mannor et al. (2007). Different from Mannor et al. (2007), we derive our results by using an implicit function theorem on the corresponding Bellman equation to obtain the gradient of $Q$, viewing the latter as the solution to the equation and as a function of $\mu_R, P$. This approach is able to generalize the results for fixed policies in Mannor et al. (2007) to the optimal value functions, and also provide distributional statements as Theorem 1 and Corollary 1 above. We also note that another potential route to obtain our results is to conduct perturbation analysis on the linear program (LP) representation of the MDP, which would also give gradient information of $V^*$ (and hence also $Q$), but using the implicit function theorem here seems sufficient.

Theorem 1 and Corollary 1 can be used immediately for statistical inference. In particular, we can construct confidence regions for subsets of the Q-value jointly, or for linear combinations of the Q-values. A quantity of interest that we will later utilize in designing good exploration policies is $Q(s, a_1) - Q(s, a_2)$, i.e. the difference between action $a_1$ and $a_2$ when the agent is in state $s$. Define $\sigma_{\Delta Q}^2$ as

$$\sigma_{\Delta Q}^2(s, a_1, a_2) = (e_{(s-1)m_a + a_1} - e_{(s-1)m_a + a_2})^T \Sigma (e_{(s-1)m_a + a_1} - e_{(s-1)m_a + a_2}) \tag{6}$$

and its estimator $\hat{\sigma}_{\Delta Q, n}^2$ by replacing $Q$, $V^*$, $\sigma_{R,n}^2$, $w$, $P$ with $\hat{Q}_n, \hat{V}_n^*$ $\hat{\sigma}_{R,n}^2$, $\hat{w}_n$, $\hat{P}_n$ in $\Sigma$, where $\hat{w}_n$ is the empirical frequency of visiting each state-action pair, i.e. $\hat{w}_n(i, j) = \frac{1}{n} \sum_{1 \le t \le n} \mathbb{1}(s_t = i, a_t = j)$. Then the $100(1 - \alpha)\%$ confidence interval (CI) for $Q(s, a_1) - Q(s, a_2)$ takes the form $\left( \hat{Q}_n(s, a_1) - \hat{Q}_n(s, a_2) \right) \pm z_\alpha \hat{\sigma}_{\Delta Q, n}^2(s, a_1, a_2)$, where $z_\alpha$ is the $(1 - \alpha/2)$-quantile of $\mathcal{N}(0, 1)$.

### 3.2 Non-Unique Optimal Policy

Suppose the optimal policy for the MDP $\mathcal{M}$ is not unique, i.e., Assumption 2 does not hold. In this situation, the estimated $\hat{Q}_n$ and $\hat{V}_n^*$ may "jump" around different optimal actions, leading to a more complicated large-sample behavior as described below:

**Theorem 2.** *Suppose Assumptions 1 and 3 hold but there is no unique optimal policy. Then there exists $K \ge 1$ distinct $m_s \times (Nm_s + N)$ matrices $\{G_k\}_{1 \le k \le K}$ and a deterministic partition of $U = \{u \in \mathcal{R}^{m_s N + m_s} : ||u|| = 1\} = \cup_{1 \le k \le K} U_k$ such that $\sqrt{n}(\hat{V}_n^* - V^*) \Rightarrow \sum_{k=1}^K G_k \mathbb{1}(Z/||Z|| \in U_k) Z$, where $Z = \mathcal{N}(0, \Sigma_{R,P})$, $\Sigma_{R,P} = Diag(W^{-1}D_R, D_P)$ and $D_P = Diag(\Sigma_{P_{1,1}}/w(0m_a + 1), \dots, \Sigma_{P_{i,j}}/w((i - 1)m_a + j), \dots, \Sigma_{P_{m_s, m_a}}/w((m_s - 1)m_a + m_a)).$*

In the case that $K > 1$ in Theorem 2, the limit distribution becomes non-Gaussian. This arises because the sensitivity to $P$ or $\mu_R$ can be very different depending on the perturbation direction, which is a consequence of solution non-uniqueness that can be formalized as a non-degeneracy in the LP representation of the MDP. We note that this phenomenon is analogous to the "non-regularity" concept in DTR that arises because the "true" parameters in these problems are very close to the decision "boundaries", which makes the obtained policy highly sensitive to estimation noises and incurs a $1/\sqrt{n}$-order bias behavior. Our case of non-unique optimal policy here captures precisely this same behavior, where we see in Theorem 2 that when $K > 1$ the asymptotic limit no longer has mean zero and consequently a $1/\sqrt{n}$-order bias arises.

We also develop two other generalizations of large-sample results, for constrained MDP and approximate value iteration respectively (see Appendices A.1 and A.2).

## 4 EFFICIENT EXPLORATION POLICY

We utilize our results in Section 3 to design exploration policies. We focus on the setting where an agent is assigned a period to collect data by running the state transition with an exploration policy. The goal is to obtain the best policy at the end of the period in a probabilistic sense, i.e., minimize the probability of selecting a suboptimal policy for the accumulated reward.

We propose a strategy that maximizes the worst-case relative discrepancy among all Q-value estimates. More precisely, we define, for $i \in \mathcal{S}$, $j \in \mathcal{A}$ and $j \neq a^*(i)$, the relative discrepancy as

$$h_{ij} = (Q(i, a^*(i)) - Q(i, j))^2 / \sigma^2_{\Delta Q}(i, a^*(i), j),$$

where $\sigma^2_{\Delta Q}(i, a^*(i), j)$ is defined in (6). Our procedure attempts to maximize the minimum of $h_{ij}$'s,

$$\max_{w \in \mathcal{W}_\eta} \min_{i \in \mathcal{S}} \min_{j \in \mathcal{A}, j \neq a^*(i)} h_{ij}, \tag{7}$$

where $w$ denotes the proportions of visits on the state-action pairs, within some allocation set $\mathcal{W}_\eta$ (which we will explain). Intuitively, $h_{ij}$ captures the relative "difficulty" in obtaining the optimal policy given the estimation errors of Q's. If the Q-values are far apart, or if the estimation variance is small, then $h_{ij}$ is large which signifies an "easy" problem, and vice versa. Criterion (7) thus aims to make the problem the "easiest". Alternatively, one can also interpret (7) from a large deviations view (Glynn & Juneja, 2004; Dong & Zhu, 2016). Suppose the Q-values for state $i$ between two different actions $a^*(i)$ and $j$ are very close. Then, one can show that the probability of suboptimal selection between the two has roughly an exponential decay rate controlled by $h_{ij}$. Obviously, there can be many more comparisons to consider, but the exponential form dictates that the smallest decay rate dominates the calculation, thus leading to the inner min's in (7). Criterion like (7) is motivated from the OCBA procedure in simulation optimization (which historically has considered simple mean-value alternatives (Chen & Lee, 2011)). Here, we consider the Q-values. For convenience, we call our procedure Q-OCBA.

Implementing criterion (7) requires two additional considerations. First, solving (7) needs the model primitives $Q$, $P$ and $\sigma^2_R$ that appear in the expression of $h_{ij}$. These quantities are unknown a priori, but as we collect data they can be sequentially estimated. This leads to a multi-stage optimization plus parameter update scheme. Second, since data are collected through running a Markov chain on the exploration actions, not all allocation $w$ is *admissible*, i.e., realizable as the stationary distribution of the MDP. To resolve this latter issue, we will derive a convenient characterization for admissibility.

Call $\pi(\cdot|s)$ admissible if the Markov Chain with transition probability $\tilde{P}^\pi$, defined for Assumption 3, is positive recurrent, and denote $w_\pi$ as its stationary distribution. Define the set $\mathcal{W} = \Big\{ w > 0 : \sum_{1 \leq j \leq m_a} w((i-1)m_a + j) = \sum_{1 \leq k \leq m_s} \sum_{1 \leq l \leq m_a} w((k-1)m_a + l)P(i|k, l)$

$\forall 1 \leq i \leq m_s, \sum_{1 \leq i \leq m_s} \sum_{1 \leq j \leq m_a} w((i-1)m_a + j) = 1 \Big\}$. The following provides a characterization of the set of admissible $\pi$:

**Lemma 1.** *For any admission policy* $\pi$, $w_\pi \in \mathcal{W}$. *For any* $w \in \mathcal{W}$, $\pi_w$ *with* $\pi_w(a = j|s = i) = w((i-1)m_a + j)/\left(\sum_{k=1}^{m_a} w((i-1)m_a + k)\right)$ *is an admissible policy.*

In other words, optimizing over the set of admissible policies is equivalent to optimizing over the set of stationary distributions. The latter is much more tractable thanks to the linear structure of $\mathcal{W}$. In practice, we will use $\mathcal{W}_\eta = \mathcal{W} \cap \{w \geq \eta\}$ for some small $\eta > 0$ to ensure closedness of the set (our experiments use $\eta = 10^{-6}$).

Algorithm 1 describes Q-OCBA. In our experiments shown next, we simply use two stages, i.e., $K = 2$. Finally, we also note that criterion like (7) can be modified according to the decision goal.

For example, if one is interested in obtaining the best estimate of $\chi^*$, then it would be more beneficial to consider $\min_{w \in \mathcal{W}_\eta} \sigma_\chi^2$. We showcase this with additional experiments in the Appendix.

---

**Input:** Number of iterations $K$, length of each batch $\{B_k\}_{1 \leq k \leq K}$, initial exploration policy $\pi_0$;
**Initialization:** $k = 0$;
**while** $k \leq K$ **do**
    Run $\pi_k$ for $B_k$ steps and set $k = k + 1$;
    Calculate $\hat{P}_{B_k}$, $\hat{\mu}_{R,B_k}$, $\hat{\sigma}_{R,B_k}^2$ and $\hat{w}_{B_k}$ based on the $B_k$ data points collected ;
    Apply value-iteration using $\hat{P}_{B_k}$ and $\hat{\mu}_{R,B_k}^2$ to obtain $\hat{Q}_{B_k}$ ;
    Plug the estimates $\hat{P}_{B_k}$, $\hat{\sigma}_{R,B_k}^2$ and $\hat{Q}_{B_k}$ into (7) to solve for the optimal $w_k$ ;
    Set $\pi_k(a = j | s = i) = w_k((i-1)m_a + j) / \sum_{l=1}^{m_a} w_k((i-1)m_a + l)$;
**end**

**Algorithm 1:** Q-OCBA sequential updating rule for exploration

---

Note that (7) is equivalent to $\min_w \max_{i \in \mathcal{S}} \max_{j \in \mathcal{A}, j \neq a^*(i)} \sum_{s,a} c_{ij}(s,a)/w_{s,a}$ subject to $w \in \mathcal{W}_\eta$, where $c_{ij}(s,a)$'s are non-negative coefficients. Based on the closed-form characterization of $\Sigma$ in Theorem 1, $c_{ij}(s,a)$'s can be estimated with plug-in estimators using data collected in earlier stages.

## 5 Numerical Experiments

We conduct several numerical experiments to support our large-sample results in Sections 3 and demonstrate the performance of Q-OCBA against some benchmark methods. We use the RiverSwim problem in (Osband et al., 2013) with $m_s$ states and two actions at each state: swim left (0) or swim right (1) (see Figure 1). The triplet above each arc represents i) the action, 0 or 1, ii) the transition probability to the next state given the current state and action, iii) the reward under the current state and action. Note that, in this problem, rewards are given only at the left and right boundary states (where the value of $r_L$ will be varied). We consider the infinite horizon setting with $\gamma = 0.95$ and $\rho = [1/m_s, \ldots, 1/m_s]^T$.

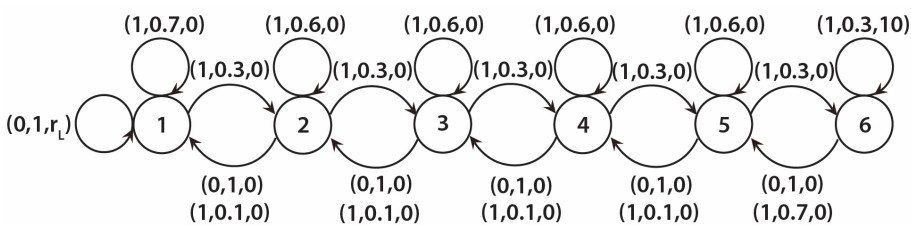

Figure 1: RiverSwim Problem

We first demonstrate the validity of our large-sample results. We use a policy that swims right with probability 0.8 at each state, i.e. $\pi(1|s) = 0.8$. Tables 1 and 2 show the coverage rates of the constructed 95% CIs, for a small $m_s = 6$ (using Theorem 1 and Corollary 1) and a large $m_s = 31$ (using Theorem 4 in the Appendix) respectively. The latter case uses a linear interpolation with $\mathcal{S}_0 = \{1, 4, \ldots, 28, 31\}$. All coverage rates are estimated using $10^3$ independent experimental repetitions (the bracketed numbers in the tables show the half-widths of 95% CI for the coverage estimates). For the Q-values, we report the average coverage rate over all $(s,a)$ pairs. When the number of observations $n$ is large enough ($\geq 3 \times 10^4$ for exact update and $\geq 10^5$ for interpolation), we see highly accurate CI coverages, i.e., close to 95%.

Table 1: Exact tabular update

| $n$ | $10^4$ | $3 \times 10^4$ | $5 \times 10^4$ |
|---|---|---|---|
| $Q$ | 0.77(0.03) | 0.93(0.02) | 0.96(0.01) |
| $\chi^{\pi^*}$ | 0.77(0.03) | 0.93(0.02) | 0.96(0.01) |

Table 2: Approximate value iteration

| $n$ | $10^4$ | $10^5$ | $10^6$ |
|---|---|---|---|
| $Q$ | 0.53(0.02) | 0.95(0.01) | 0.95(0.01) |
| $\chi^{\pi^*}$ | 0.80(0.03) | 0.94(0.02) | 0.95(0.01) |

Next we investigate the efficiency of our exploration policy. We compare Q-OCBA with $K = 2$ to four benchmark policies: i) $\epsilon$-greedy with different values of $\epsilon$, ii) random exploration (RE) with different values of $\pi(1|s)$, iii) UCRL2 (a variant of UCRL) with $\delta = 0.05$ (Jaksch et al., 2010), iv) PSRL with different posterior updating frequencies (Osband et al., 2013), i.e., PSRL($x$) means PSRL is implemented with $x$ episodes. We use $m_s = 6$ and vary $r_L$ from 1 to 3. To ensure fairness, we use a two-stage implementation for all policies, with 30% of iterations first dedicated to RE (with $\pi(1|s) = 0.6$) as a warm start, i.e., the data are used to estimate the parameters needed for the second stage. To give enough benefit of the doubt, we notice the probabilities of correct selection for both UCRL2 and PSRL are much worse without the warm start.

Tables 3 and 4 compare the probabilities of obtaining the optimal policy (based on the estimated $\hat{Q}_n$'s). For $\epsilon$-greedy, RE, and PSRL, we report the results with the parameters that give the best performances in our numerical experiments. The probability of correct selection is estimated using $10^3$ replications of the procedure. We observe that Q-OCBA substantially outperforms the other methods, both with a small data size ($n = 10^3$ in Table 3) and a larger one ($n = 10^4$ in Table 4). Generally, these benchmark policies perform worse for larger values of $r_L$. This is because for small $r_L$, the $(s, a)$ pairs that need to be explored more also tend to have larger $Q$-values. However, as $r_L$ increase, there is a misalignment between the $Q$-values and the $(s, a)$ pairs that need more exploration.

The superiority of our Q-OCBA in these experiments come as no surprise to us. The benchmark methods like UCRL2 and PSRL are designed to minimize regret which involves balancing the exploration-exploitation trade-off. On the other hand, Q-OCBA focuses on efficient exploration only, i.e., our goal is to minimize the probability of incorrect policy selection, and this is achieved by carefully utilizing the variance information gathered from the first stage that is made possible by our derived asymptotic formulas. We provide additional numerical results in Appendix B.

Table 3: Probability of correct selection for different exploration policies, $n = 10^3$

| $r_L$ | 0.2-greedy | RE(0.6) | UCRL2 | PSRL(100) | Q-OCBA |
|---|---|---|---|---|---|
| 1 | 0.95(0.01) | 0.70(0.03) | 0.44(0.03) | 0.53(0.03) | 0.87(0.02) |
| 2 | 0.15(0.02) | 0.29(0.03) | 0.11(0.02) | 0.33(0.03) | 0.55(0.03) |
| 3 | 0.00(0.00) | 0.45(0.03) | 0.21(0.02) | 0.41(0.03) | 0.84(0.02) |

Table 4: Probability of correct selection for different exploration policies, $n = 10^4$

| $r_L$ | 0.2-greedy | RE(0.6) | UCRL2 | PSRL(100) | Q-OCBA |
|---|---|---|---|---|---|
| 1 | 1.00(0.00) | 0.95(0.01) | 0.82(0.02) | 1.00(0.00) | 1.00(0.00) |
| 2 | 0.55(0.03) | 0.80(0.03) | 0.52(0.03) | 0.94(0.02) | 1.00(0.00) |
| 3 | 0.21(0.03) | 0.94(0.01) | 0.75(0.03) | 0.76(0.03) | 1.00(0.00) |

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

# A    ADDITIONAL THEORETICAL RESULTS

In this section, we present additional results on large-sample behaviors for constrained MDPs and also estimations based on approximation value iteration.

## A.1    CONSTRAINED PROBLEMS

We consider the constrained MDP setting for budgeted decision-making (Boutilier & Lu, 2016) and more recently safety-critical applications (Achiam et al., 2017; Chow et al., 2017). Suppose now we aim to maximize the long-run accumulated discounted reward, $V^\pi(s) = \mathbb{E}[\sum_{t=0}^{\infty} \gamma^t R(s_t, \pi(s_t))|s_0 = s]$, while at the same time want to ensure that a long-run accumulated discounted cost, denoted as $L^\pi(s) = \mathbb{E}[\sum_{t=0}^{\infty} \gamma^t C(s_t, \pi(s_t))|s_0 = s]$ which we call the loss function, is constrained by some given value $\eta$, i.e.,

$$\max_{\pi} \sum_s \rho(s)V^\pi(s) \text{ subject to } \sum_s \rho(s)L^\pi(s) \leq \eta \tag{8}$$

We assume data coming in like before and, in addition, that we have observations on the incurred cost at each sample of $(s, a)$. Call the empirical estimate of the cost $\hat{\mu}_{C,n}$. We follow our paradigm to solve the empirical counterpart of the problem, namely to find a policy $\hat{\pi}_n^*$ that solves (8) by using $\hat{V}_n^\pi(s)$ and $\hat{L}_n^\pi(s)$ instead of $V^\pi(s)$ and $L^\pi(s)$, where $\hat{V}_n^\pi(s)$'s and $\hat{L}_n^\pi(s)$'s are the value functions and loss functions evaluated using the empirical estimates $\hat{\mu}_{R,n}, \hat{\mu}_{C,n}, \hat{P}_n$. We focus on the estimation error of the optimal value (instead of the feasibility, which could also be important but not pursued here).

To understand the error, we first utilize an optimality characterization of constrained MDPs. In general, an optimal policy for (8) is a "split" policy (Feinberg & Rothblum, 2012), namely, a policy that is deterministic except that at one particular state a randomization between two different actions is allowed. This characterization can be deduced from the associated LP using occupancy measures (Altman, 1999). We call the randomization probability the mixing parameter $\alpha^*$, i.e., whenever this particular state, say $s_r$, is visited, action $a_1^*(s_r)$ is chosen with probability $\alpha^*$ and action $a_2^*(s_r)$ is chosen with probability $1 - \alpha^*$. We then have the following result:

**Theorem 3.** *Suppose Assumptions 1 and 3 hold and there is a unique optimal policy. Moreover, assume that there is no deterministic policy $\pi$ that satisfies $\sum_s \rho(s)L^\pi(s) = \eta$. Then we have $\sqrt{n}(\hat{V}_n^* - V^*) \Rightarrow N(0, \Sigma)$ as $n \to \infty$, where one of the following cases hold:*

**1.** *The optimal policy is deterministic. We then have* $\Sigma = \Sigma_V$ *where* $\Sigma_V$ *is defined in Theorem 1.*

**2.** *The optimal policy is deterministic, except at one state where a randomization between two actions occurs, with the mixing parameter* $\alpha^*$. *Denote the state where the randomization occurs by* $s_r$ *and two possible actions for* $s_r$ *by* $a_1^*(s_r)$ *and* $a_2^*(s_r)$, *We have*

$$\Sigma = \left((I - \gamma P^{\pi^*})^{-1}[G^{\pi^*}, 0, H_V^{\pi^*}] - \frac{(I - \gamma P^{\pi^*})^{-1} q_V \rho^T (I - \gamma P^{\pi^*})^{-1}[0, G^{\pi^*}, H_L^{\pi^*}]}{\rho^T (I - \gamma P^{\pi^*})^{-1} q_L}\right)$$

$$\Sigma_{R,C,P} \left((I - \gamma P^{\pi^*})^{-1}[G^{\pi^*}, 0, H_V^{\pi^*}] - \frac{(I - \gamma P^{\pi^*})^{-1} q_V \rho^T (I - \gamma P^{\pi^*})^{-1}[0, G^{\pi^*}, H_L^{\pi^*}]}{\rho^T (I - \gamma P^{\pi^*})^{-1} q_L}\right)^T,$$

*where* $\Sigma_{R,C,P} = Diag(W^{-1}D_R, W^{-1}D_C, D_P)$, $q_V$ *and* $q_L$ *are* $m_s$-*dimensional vectors with* $q_V(s) = (\mu_R(s, a_1^*(s)) - \mu_R(s, a_2^*(s))) + \sum_{j=1}^{m_s} \gamma V^{\pi^*}(j)(P(j|s, a_1^*(s)) - P(j|s, a_2^*(s)))$ *and* $q_L(s) = (\mu_C(s, a_1^*(s)) - \mu_C(s, a_2^*(s))) + \sum_{j=1}^{m_s} \gamma L^{\pi^*}(j)(P(j|s, a_1^*(s)) - P(j|s, a_2^*(s)))$ *when* $s = s_r$, *and* $q_v(s) = q_L(s) = 0$ *when* $s \neq s_r$,

$$G^{\pi^*} = \begin{pmatrix} \pi^*(\cdot|1)^T & & \\ & \ddots & \\ & & \pi^*(\cdot|m_s)^T \end{pmatrix}, H_V^{\pi^*} = \begin{pmatrix} q_V^{\pi^*}(1)^T & & \\ & \ddots & \\ & & q_V^{\pi^*}(m_s)^T \end{pmatrix}$$

*where* $q_V^{\pi^*}(i)^T = \gamma[\pi^*(1|i)(V^{\pi^*})^T, \ldots, \pi^*(j|i)(V^{\pi^*})^T, \ldots, \pi^*(m_a|i)(V^{\pi^*})^T]$, *which is an* $N$-*dimensional row vector, and* $H_L^{\pi^*}$ *and* $q_L^{\pi^*}(i)$ *are defined similarly by substituting* $V$ *with* $L$.

Case 1 in Theorem 3 corresponds to the case where the constraint in (8) is non-binding. This effectively reduces to the unconstrained scenario in Theorem 1 since a small perturbation of $\mu_R, \mu_C, P$ does not affect feasibility. Case 2 is when the constraint is binding. In this case, $\alpha^*$ must be chosen such that the split policy ensures equality in the constraint, and when $\mu_R, \mu_C, P$ is perturbed the estimated $\hat{\alpha}_n^*$ would adjust accordingly. Thus, in this case the estimation of $V^*$ incurs two sources of noises, one from the uncertainty of $R, P$ that appears also in unconstrained problems, and one from the uncertainty in calibrating $\hat{\alpha}_n^*$ that is in turn affected by $C, P$, thus leading to the extra terms in the variance expression.

## A.2 Approximate Value Iteration

When the state space $\mathcal{S}$ is large, updating an $m_s \times m_a$ look-up table via $\mathcal{T}_{\mu_R,P}(.)$ can be computationally infeasible. Approximate value iteration operates by applying a mapping $M$ over $\mathcal{T}_{\mu_R,P}$. In many cases, $M = M_g \circ M_I^{S_0}$ where $M_I^{S_0}$ is a dimension-reducing "inherit" mapping $\mathbb{R}^{m_s m_a} \to \mathbb{R}^{m_{s_0} m_a}$, and $M_g$ is the "generalization" mapping $\mathbb{R}^{m_{s_0} m_a} \to \mathbb{R}^{m_s m_a}$ that lifts back to the full dimension. By selecting a "representative" subset $\mathcal{S}_0 \subset \mathcal{S}$ with cardinality $|\mathcal{S}_0| = m_{s_0} \ll m_s$, $M_I^{S_0}$ is defined as $M_I^{S_0}(x) = [x(i, j)]_{i \in \mathcal{S}_0, 1 \leq j \leq m_a}$ where $[x_i]_{i \in I}$ denotes the set of entries of $x$ whose index $i \in I$. In this setup, we define $Q^M$ as a fixed point of the operator $M \circ \mathcal{T}_{\mu_R,P}(\cdot)$, and $V^M(s) = \max_a Q^M(s, a)$. We also define $Q_{S_0}^M = M_I^{S_0} \circ \mathcal{T}_{\mu_R,P}(Q^M)$ as the dimension-reduced Q-value.

We derive large-sample error estimates in this case. For this, we first assume there is a well-defined metric on $\mathcal{S}$. To guarantee the existence of $Q^M$, we make the following assumption on the generalization map $M_g$:

**Assumption 4.** *$M_g$ is a max-norm non-expansion mapping in $\mathcal{S}$, i.e.,* $||M_g(x) - M_g(y)||_\infty < ||x - y||_\infty \quad \forall x, y \in \mathcal{S}$.

We also need the following analogs of Assumptions 2 and 3 to $Q^M$ and $\mathcal{S}_0$:

**Assumption 5.** *For any state $s \in \mathcal{S}$, $\arg\max_{a \in \mathcal{A}} Q^M(s, a)$ is unique.*

**Assumption 6.** *For the Markov Chain with transition probability $\tilde{P}^\pi$, the set of states $\{(s, a) : s \in \mathcal{S}_0, a \in \mathcal{A}\}$ is in the same communication class and this class is positive recurrent.*

Let $N_0 = m_{s_0} m_a$ and $I_{\mathcal{S}_0} = \{(i-1)m_a + j : i \in \mathcal{S}_0, a \in \mathcal{A}\}$. With Assumption 6, we denote $\tilde{P}^M_{\mathcal{S}_0}$ as a sub-matrix of the matrix $\tilde{P}^\pi$ that only contains rows with indexes in $I_{\mathcal{S}_0}$. We also denote $\mathcal{S}_0(i)$ as the $i$-th element (state) in $\mathcal{S}_0$. We define $\hat{Q}^M_n$ as the empirical estimator of $Q^M$ built on $n$ observations. Then we have:

**Theorem 4.** *Under Assumption 4, 5 and 6, if $M_g$ is continuously differentiable, then $\sqrt{n}(\hat{Q}^M_n - Q^M) \Rightarrow \mathcal{N}(0, \Sigma^M_{\mathcal{S}_0})$ as $n \to \infty$, where*

$$\Sigma^M_{\mathcal{S}_0} = (I - \gamma \nabla M_g(Q^M_{\mathcal{S}_0})\tilde{P}^M_{\mathcal{S}_0})^{-1} \nabla M_g(Q^M_{\mathcal{S}_0})(W^{\mathcal{S}_0})^{-1}[D^{\mathcal{S}_0}_R + D^{\mathcal{S}_0}_Q]$$
$$\nabla M_g(Q^M_{\mathcal{S}_0})^T((I - \gamma \nabla M_g(Q^M_{\mathcal{S}_0})\tilde{P}^M_{\mathcal{S}_0})^{-1})^T,$$

*$\nabla M_g$ is the Jacobian of mapping $M_g$, $W^{\mathcal{S}_0}, D^{\mathcal{S}_0}_R, D^{\mathcal{S}_0}_Q$ are $N_0 \times N_0$ diagonal matrices with $W^{\mathcal{S}_0}((i-1)m_a + j, (i-1)m_a + j) = w(\mathcal{S}_0(i), j)$, $D^{\mathcal{S}_0}_R((i-1)m_a + j, (i-1)m_a + j) = \sigma^2_R(\mathcal{S}_0(i), j)$, $D^{\mathcal{S}_0}_Q((i-1)m_a + j, (i-1)m_a + j) = (V^M)^T \Sigma_{P_{\mathcal{S}_0(i),j}} V^M$.*

Assumption 4 is generally satisfied by "local" approximation methods such as linear interpolation, $k$-nearest neighbors and local weighted average (Gordon, 1995). In all these cases, $\nabla M_g$ in Theorem 4 is actually a constant matrix.

# B  ADDITIONAL NUMERICAL RESULTS

This section reports additional numerical experiments. Sections B.1 present further results on the estimation quality of Q-values, $V^*$ and $\chi^*$. Section B.2 provides additional results to demonstrate the efficiency of our proposed exploration strategy.

## B.1  STATISTICAL QUALITY OF INTERVAL ESTIMATORS

In this section, we provide additional numerical results about Tables 1 and 2 in the main paper. For Q-values in Table 1 of main paper, we only report the average coverage rate over all $(s, a)$ pairs. Table 5 presents the coverage rates for each individual Q-values. It also provides the coverage rates for the value functions and $\chi^*$, where $\chi^*$ is the uniformly initialized value function, i.e. $\rho = [1/m_s, \ldots, 1/m_s]^T$. We use RE with $\pi(1|s) = 0.5$ as our exploration policy. We see that the behaviors of these individual estimates are largely consistent. The coverage rates all converge to the nominal 95% as the number of observations $n$ increases. Moreover, the coverages for the individual $Q$'s, $V^*$'s, and the averages of these quantities are similar at any given $n$. Specifically, when $n = 10^4$, the coverages are all around $77\% - 78\%$, when $n = 3 \times 10^4$ they are all around 93%, and when $n = 5 \times 10^4$ they are all very close to 95%. These suggest a sample size of $5 \times 10^4$ (or lower) is enough to elicit our asymptotic results in Theorem 1 and Corollary 1 in this problem.

Table 5: Coverage for $Q(s,a)$, $V^*$ and $\chi^*$ values using exact tabular update

| $n$ | $10^4$ | $3 \times 10^4$ | $5 \times 10^4$ |
|---|---|---|---|
| $Q(1,0)$ | 0.773(0.026) | 0.930(0.016) | 0.956(0.013) |
| $Q(1,1)$ | 0.773(0.026) | 0.930(0.016) | 0.956(0.013) |
| $Q(2,0)$ | 0.773(0.026) | 0.930(0.016) | 0.956(0.013) |
| $Q(2,1)$ | 0.771(0.026) | 0.930(0.016) | 0.955(0.013) |
| $Q(3,0)$ | 0.771(0.026) | 0.930(0.016) | 0.955(0.013) |
| $Q(3,1)$ | 0.771(0.026) | 0.929(0.016) | 0.957(0.013) |
| $Q(4,0)$ | 0.771(0.026) | 0.929(0.016) | 0.957(0.012) |
| $Q(4,1)$ | 0.775(0.026) | 0.934(0.015) | 0.958(0.012) |
| $Q(5,0)$ | 0.775(0.026) | 0.934(0.015) | 0.958(0.012) |
| $Q(5,1)$ | 0.775(0.026) | 0.935(0.015) | 0.954(0.013) |
| $Q(6,0)$ | 0.775(0.026) | 0.935(0.015) | 0.954(0.013) |
| $Q(6,1)$ | 0.750(0.026) | 0.920(0.017) | 0.950(0.014) |
| Average of $Q$ | 0.771(0.026) | 0.931(0.016) | 0.956(0.013) |
| $V^*(1)$ | 0.773(0.026) | 0.930(0.016) | 0.956(0.013) |
| $V^*(2)$ | 0.772(0.026) | 0.930(0.016) | 0.955(0.013) |
| $V^*(3)$ | 0.774(0.026) | 0.929(0.016) | 0.957(0.013) |
| $V^*(4)$ | 0.776(0.026) | 0.934(0.015) | 0.958(0.012) |
| $V^*(5)$ | 0.772(0.026) | 0.935(0.015) | 0.954(0.013) |
| $V^*(6)$ | 0.752(0.027) | 0.920(0.017) | 0.950(0.014) |
| Average of $V^*$ | 0.769(0.026) | 0.929(0.016) | 0.955(0.012) |
| $\chi^*$ | 0.772(0.026) | 0.933(0.015) | 0.958(0.012) |

Tables 6, 7 and 8 compare the CI coverage rates when the state space is large, i.e. $m_s = 31$, using RE with different values of $\pi(1|s)$, i.e., $\pi(1|s) = 0.8, 0.85$, and 0.9. Compared to exact update, the coverage convergence for approximate update appears generally slower. Specifically, comparing Tables 5 and 6 that use the same RE with $\pi(1|s) = 0.8$, we see that the coverages on the averages of $Q$'s and $V^*$'s for approximate update are only around $23\% - 25\%$ when $n = 10^4$, whereas they are $77\% - 78\%$ for exact update. Also, while the nominal coverage, i.e., 95%, is obtained when $n = 5 \times 10^4$ in the exact update for all studied quantities, this sample size is not enough for approximate update, where it appears we need $n$ to be of order $10^7$ to obtain the nominal coverage.

Furthermore, Tables 6, 7 and 8 show that the rates of convergence to the nominal coverage are quite different for different values of $\pi(1|s)$'s. The convergence rate when $\pi(1|s) = 0.85$ seems to be the fastest, with the coverage close to 95% already when $n = 10^5$. On the other hand, when $\pi(1|s) = 0.8$, the coverage is close to 95% only when $n$ is as large as $10^7$, and when $\pi(1|s) = 0.9$, even $n = 10^7$ is not enough for convergence to kick in. We also see that, when the coverage is very far from the nominal rate, discrepancies can show up among the estimates of $Q$, $V^*$ and $\chi^*$. For example, when $\pi(1|s) = 0.8$ and $n = 10^4$, the coverages of $Q$ and $V^*$ are around $23\% - 25\%$ but the coverage of $\chi^*$ is as low as 1%, and when $\pi(1|s) = 0.9$ and $n = 10^7$, the coverages of $Q$ and $V^*$ are around $75\% - 77\%$ but that of $\chi^*$ is only 29%. However, in settings where the coverage is close to 95%, all these quantities appear to attain this accuracy simultaneously in all considered cases. These caution that coverage accuracy can be quite sensitive to the specifications of the exploration policy. Nonetheless, the convergence behaviors predicted by Theorem 1, Corollary 1 and Theorem 4 are all observed to hold.

Table 6: Linear interpolation in approximate value iteration with $\pi(1|s) = 0.8$

| $n$ | $10^4$ | $10^5$ | $10^6$ | $10^7$ |
|---|---|---|---|---|
| Average of $Q$ | 0.23(0.01) | 0.55(0.03) | 0.92(0.02) | 0.95(0.01) |
| Average of $V^*$ | 0.25(0.01) | 0.56(0.02) | 0.93(0.01) | 0.95(0.01) |
| $\chi^*$ coverage | 0.01(0.01) | 0.35(0.03) | 0.94(0.02) | 0.94(0.02) |

Table 7: Linear interpolation in approximate value iteration with $\pi(1|s) = 0.85$

| $n$ | $10^4$ | $10^5$ | $10^6$ | $10^7$ |
|---|---|---|---|---|
| Average of $Q$ | 0.53(0.02) | 0.95(0.01) | 0.95(0.01) | 0.95(0.01) |
| Average of $V^*$ | 0.59(0.02) | 0.95(0.01) | 0.95(0.01) | 0.95(0.01) |
| $\chi^*$ coverage | 0.80(0.03) | 0.94(0.02) | 0.95(0.01) | 0.94(0.01) |

Table 8: Linear interpolation in approximate value iteration with $\pi(1|s) = 0.9$

| $n$ | $10^4$ | $10^5$ | $10^6$ | $10^7$ |
|---|---|---|---|---|
| Average of $Q$ | 0.31(0.01) | 0.49(0.01) | 0.67(0.01) | 0.75(0.02) |
| Average of $V^*$ | 0.37(0.01) | 0.54(0.01) | 0.71(0.01) | 0.77(0.02) |
| $\chi^*$ coverage | 0.25(0.03) | 0.30(0.03) | 0.28(0.02) | 0.29(0.03) |

## B.2 EFFICIENCY OF EXPLORATION POLICIES

In Q-OCBA, our second-stage exploration policy is derived by maximizing the worst-case relative discrepancy among all Q-value estimates. If one is interested in obtaining the best estimate of $\chi^*$ (i.e., the optimal value function initialized at a distribution $\rho$), then it would be more beneficial to consider solving

$$\min_{w \in \mathcal{W}_\eta} \sigma_\chi^2 \tag{9}$$

to derive the optimal second-stage exploration policy $\pi_w$ (recall Lemma 1). The motivation is that by doing so we would obtain a CI for $\chi^*$ as short as possible.

Table 9 compares the 95% CI lengths and coverages for this exploration policy with other benchmark strategies, for $r_L$ ranging from 1 to 3. For each $r_L$, we show the averages of the coverages and CI lengths of $Q$ estimates among all $(s, a)$ pairs, and also the coverage and CI length of $\chi^*$ estimates. Note that our strategy intends to shorten the CI lengths of $\chi^*$ estimates. Like our experiment in Section 5 in the main paper, we use a total observation budget $n = 10^4$, and devote 30% to the initial stage where RE with $\pi(1|s) = 0.8$ is used to estimate the parameters used to plug in the criterion to be optimized in the second stage. For convenience and the consistency of terminology, we continue to call our procedure to attain criterion (9) Q-OCBA. We compare this with pure RE and $\epsilon$-greedy, with $\epsilon$ ranging from 0.01 to 0.2.

Table 9 shows that our budget is enough to achieve the nominal 95% coverages for both the Q-values and $\chi^*$ using all strategies, which is consistent with the conclusion from Theorem 1 and Corollary 1. However, Q-OCBA leads to desirably much shorter CI's generally, with the shortest CI lengths in all settings and sometimes by a big margin. For example, when $r_L = 2$, the CI length derived by Q-OCBA is at least 80% less than those derived by all the other methods. We also observe that Q-OCBA performs much more stably than RE and $\epsilon$-greedy, the latter varying quite significantly

for different values of $r_L$. When $r_L = 1$, $\epsilon$-greedy with $\epsilon = 0.01$ can perform almost as well as Q-OCBA, with both CI lengths for $Q$ being $2.45 - 2.46$ and for $\chi^*$ being $2.41 - 2.42$. But when $r_L = 2$, $\epsilon$-greedy with the same $\epsilon = 0.01$ cannot even explore all $(s, a)$ pairs. The situation worsens when $r_L = 3$, where none of the considered values of $\epsilon$ can explore all $(s, a)$ pairs. This observation on $\epsilon$-greedy is consistent with Table 3 in the main paper where we consider the criterion using the probability of correct selection. Regardless of using that or the current criteria, the performances of $\epsilon$-greedy depend fundamentally on whether the $(s, a)$ pairs that need to be explored more also tend to have larger Q-values. Note that when changing $r_L$, the corresponding changes in the Q-values would change the exploration "preference" for $\epsilon$-greedy. However, as the underlying stochasticity of the system does not change with $r_L$, the states that need more exploration remain unchanged. This misalignment leads us to observe quite different performances for $\epsilon$-greedy when $r_L$ varies. Lastly, again consistent with the results on the probability of correct selection shown in Table 3 of the main paper, we observe that Q-OCBA outperforms pure RE in all cases in Table 5, with at least 40% shorter CI lengths for the $\chi^*$ estimates. This is attributed to the efficient use of variance information in the second stage of Q-OCBA.

Table 9: Length of CI comparison for different exploration policies

|  | $\epsilon = 0.2$ | $\epsilon = 0.1$ | $\epsilon = 0.01$ | $\pi(1|s) = 0.8$ | Q-OCBA |
|---|---|---|---|---|---|
| $r_L = 1$ | | | | | |
| $Q$ Coverage | 0.94(0.05) | 0.97(0.03) | 0.97(0.03) | 0.97(0.01) | 0.96(0.04) |
| $Q$ CI length | 3.86(0.16) | 2.73(0.09) | 2.46(0.03) | 3.85(0.05) | 2.45(0.03) |
| $\chi^*$ Coverage | 0.94(0.05) | 0.97(0.03) | 0.98(0.03) | 0.97(0.01) | 0.95(0.04) |
| $\chi^*$ CI length | 4.11(0.04) | 2.86(0.01) | 2.42(0.01) | 4.10(0.01) | 2.41(0.01) |
| $r_L = 2$ | | | | | |
| $Q$ Coverage | 0.98(0.01) | 0.96(0.01) | NA [a] | 0.97(0.01) | 0.97(0.01) |
| $Q$ CI length | 2.25(0.14) | 2.72(0.17) | NA | 1.84(0.11) | 0.32(0.02) |
| $\chi^*$ coverage | 0.96(0.01) | 0.94(0.02) | NA | 0.96(0.01) | 0.96(0.01) |
| $\chi^*$ CI length | 2.69(0.02) | 3.23(0.04) | NA | 2.20(0.01) | 0.37(0.01) |
| $r_L = 3$ | | | | | |
| $Q$ coverage | NA | NA | NA | 0.97(0.01) | 0.97(0.01) |
| $Q$ CI length | NA | NA | NA | 0.74(0.06) | 0.40(0.03) |
| $\chi^*$ coverage | NA | NA | NA | 0.95(0.01) | 0.96(0.01) |
| $\chi^*$ CI length | NA | NA | NA | 0.91(0.01) | 0.49(0.01) |

[a]NA means that some $(s, a)$ pair has never been visited.

## C   PROOFS OF MAIN RESULTS

In this section, we present the proofs of the main results. In the proofs, we shall treat $P$ as an $Nm_s$-dimensional vector following the index rule: $P((i - 1)N + (j - 1)m_s + k) = P(k|i, j)$

*Proof of Theorem 1.* Define $F(Q', r', P')$ as a mapping from $\mathbb{R}^N \times \mathbb{R}^N \times \mathbb{R}^{Nm_s}$ to $\mathbb{R}^N$. Specifically,

$$F(Q', r', P')((i - 1)m_a + j) = Q'((i - 1)m_a + j) - r'((i - 1)m_a + j)$$
$$-\gamma \sum_{1 \leq k \leq m_s} P'((i - 1)N + (j - 1)m_s + k) \, g_k(Q')$$

for $1 \leq i \leq m_s$ and $1 \leq j \leq m_a$, where $g_k(Q') = \max_l Q'((k - 1)m_a + l)$, for $1 \leq k \leq m_s$.

By Assumption 2, there exists an open neighborhood of $Q$, which we denote as $\Omega$, such that $\forall Q' \in \Omega$, $\arg\max_j Q'((i-1)m_a + j)$ is still unique for each $1 \le i \le m_s$. Then, for each $1 \le k \le m_s$, $g_k(Q')$ has all its partial derivatives exist and continuous. This implies that $F(Q', r', P')$ is continuously differentiable in $\Omega \times \mathbb{R}^N \times \mathbb{R}^{Nm_s}$.

Denote the partial derivatives of $F$ as

$$\frac{\partial F}{\partial(Q', r', P')} = \left[ \frac{\partial F}{\partial Q'} \middle| \frac{\partial F}{\partial r'} \middle| \frac{\partial F}{\partial P'} \right].$$

Note that $\frac{\partial F}{\partial Q'}$ is an $N \times N$ matrix. Denote its element at the $((i-1)m_a + j)$-th row, $((k-1)m_a + l)$-th column by $\frac{\partial F_{(i-1)m_a + j}}{\partial Q'_{(k-1)m_a + l}}$. Then we have

$$\frac{\partial F_{(i-1)m_a + j}}{\partial Q'_{(k-1)m_a + l}} = \mathbb{1}(k = i, j = l) - \gamma P'((i-1)N + (j-1)m_s + k)$$

$$\times \mathbb{1}\left( Q'((k-1)m_a + l) = \max_u Q'((k-1)m_a + u) \right).$$

Putting all the elements together, we have

$$\frac{\partial F}{\partial Q'} = I - \gamma \tilde{P}',$$

where $\tilde{P}'$ is an $N \times N$ matrix with

$$\tilde{P}'((i-1)m_a + j, (k-1)m_a + l) = P'((i-1)N + (j-1)m_s + k)$$
$$\times \mathbb{1}(Q'((k-1)m_a + l) = \max_u Q'((k-1)m_a + u)),$$

for $1 \le i \le m_s$, $1 \le j \le m_a$, $1 \le k \le m_s$, $1 \le l \le m_a$.

Since all rows of $\tilde{P}'$ sum up to one, $\tilde{P}'$ can be interpreted as the transition matrix of a Markov Chain with state space $\{(i,j) : 1 \le i \le m_s, 1 \le j \le m_a\}$. Note that $\frac{\partial F}{\partial Q'}$ is invertible for any $Q' \in \Omega$.

We can then apply the implicit function theorem to the equation $F(Q, \mu_R, P) = 0$. In particular, there exists an open set $U$ around $\mu_R \times P \in \mathbb{R}^N \times \mathbb{R}^{Nm_s}$, and a unique continuously differentiable function $\phi \colon U \to \mathbb{R}^N$, such that for any $r' \times P' \in U$

$$\phi(\mu_R, P) = Q$$
$$F(\phi(r', P'), r', P') = 0.$$

In addition, the partial derivatives of $\phi$ satisfy

$$\nabla \phi(\mu_R, P) := \left. \frac{\partial \phi}{\partial(r', P')} \right|_{r' = \mu_R, P' = P} = - \left[ \frac{\partial F}{\partial Q'} \right]^{-1} \left[ \frac{\partial F}{\partial r'}, \frac{\partial F}{\partial P'} \right] \Bigg|_{Q' = Q, r' = \mu_R, P' = P}$$

It is also easy to verify that

$$\left. \frac{\partial F}{\partial r'} \right|_{Q' = Q, r' = \mu_R, P' = P} = I_{N \times N}.$$

We also note that

$$\left. \frac{\partial F_{(i-1)m_a + j}}{\partial P'_{(k-1)N + (l-1)m_s + v}} \right|_{Q' = Q, r' = \mu_R, P' = P} = \gamma \max_u Q((v-1)m_a + u)\mathbb{1}(k = i, j = l)$$

$$= \gamma V^*(v)\mathbb{1}(k = i, j = l),$$

for $1 \leq i \leq m_s$, $1 \leq j \leq m_a$, $1 \leq k \leq m_s$, $1 \leq l \leq m_a$, and $1 \leq v \leq m_s$. Then

$$
C^{\pi^*} := \left. \frac{\partial F}{\partial P'} \right|_{Q'=Q, r'=\mu_R, P'=P} = \begin{pmatrix} (V^*)^T & & & & \\ & \ddots & & & \\ & & (V^*)^T & & \\ & & & \ddots & \\ & & & & (V^*)^T \end{pmatrix},
$$

which is an $N \times N m_s$ matrix.

Next, for

$$
\hat{\mu}_{R,n}((i-1)m_a + j) = \frac{\sum_{1 \leq t \leq n} r_t(s_t, a_t) \mathbb{1}(s_t = i, a_t = j)}{\sum_{1 \leq t \leq n} \mathbb{1}(s_t = i, a_t = j)}
$$

$$
= \frac{\sum_{1 \leq t \leq n} r_t(s_t, a_t) \mathbb{1}(s_t = i, a_t = j)}{w((i-1)m_a + j)n} \frac{w((i-1)m_a + j)n}{\sum_{1 \leq t \leq n} \mathbb{1}(s_t = i, a_t = j)}
$$

and

$$
\hat{P}_n((i-1)N + (j-1)m_s + k)
$$
$$
= \frac{\sum_{1 \leq t \leq n} \mathbb{1}(s_t = i, a_t = j, s'_t(s_t, a_t) = k)}{\sum_{1 \leq t \leq n} \mathbb{1}(s_t = i, a_t = j)}
$$
$$
= \frac{\sum_{1 \leq t \leq n} \mathbb{1}(s_t = i, a_t = j, s'_t(s_t, a_t) = k)}{w((i-1)m_a + j)n} \frac{w((i-1)m_a + j)n}{\sum_{1 \leq t \leq n} \mathbb{1}(s_t = i, a_t = j)}
$$

by Assumption 1, 3 and Slutsky's theorem, we have

$$
[\hat{\mu}_{R,n}, \hat{P}_n] - [\mu_R, P] \longrightarrow 0 \quad \text{a.s}
$$

and

$$
\sqrt{n}([\hat{\mu}_{R,n}, \hat{P}_n] - [\mu_R, P]) \Rightarrow \mathcal{N}(0, \Sigma_{R,P}) \tag{10}
$$

where $\Sigma_{R,P} = \begin{pmatrix} W^{-1}D_R & \mathbf{0} \\ \mathbf{0} & D_P \end{pmatrix}$, and

$$
D_P = \begin{pmatrix} \frac{\Sigma_{P_{1,1}}}{w(0m_a+1)} & & & & \\ & \ddots & & & \\ & & \frac{\Sigma_{P_{i,j}}}{w((i-1)m_a+j)} & & \\ & & & \ddots & \\ & & & & \frac{\Sigma_{P_{m_s,m_a}}}{w((m_s-1)m_a+m_a))} \end{pmatrix},
$$

which is an $N m_s \times N m_s$ matrix. By the continuous mapping theorem, we have

$$
\phi(\hat{\mu}_{R,n}, \hat{P}_n) - \phi(\mu_R, P) \to 0 \text{ a.s. as } n \to \infty,
$$

which implies $\hat{Q}_n \to Q$ a.s.. In addition, using the delta method, we have

$$
\sqrt{n}(\hat{Q}_n - Q) = \sqrt{n}(\phi(\hat{\mu}_{R,n}, \hat{P}_n) - \phi(\mu_R, P))
$$
$$
\Rightarrow \mathcal{N}(0, \nabla\phi(\mu_R, P)\Sigma_{R,P}\nabla\phi(\mu_R, P)^T) \text{ as } n \to \infty.
$$

We also have

$$\nabla\phi(\mu_R, P)\Sigma_{R,P}\nabla\phi(\mu_R, P)^T$$

$$= (I - \gamma\tilde{P}^{\pi^*})^{-1}[I, C^{\pi^*}]\left(\begin{array}{c|c} W^{-1}D_R & \mathbf{0} \\ \hline \mathbf{0} & D_P \end{array}\right)[I, C^{\pi^*}]^T((I - \gamma\tilde{P}^{\pi^*})^{-1})^T$$

$$= (I - \gamma\tilde{P}^{\pi^*})^{-1}(W^{-1}D_R + C^{\pi^*}D_P(C^{\pi^*})^T)((I - \gamma\tilde{P}^{\pi^*})^{-1})^T$$

$$= (I - \gamma\tilde{P}^{\pi^*})^{-1}W^{-1}[D_R + D_Q]((I - \gamma\tilde{P}^{\pi^*})^{-1})^T.$$

$\square$

*Proof of Corollary 1.* Define $g_V(Q)$: $\mathbb{R}^N \to \mathbb{R}^{m_s}$ as

$$g_V(Q) = (g_1(Q), \ldots, g_{m_s}(Q)) = (V^{\pi^*}(1), \ldots, V^{\pi^*}(m_s)),$$

which is continuously differentiable in an open neighborhood of $Q$. Then we can apply the delta method to get

$$\sqrt{n}(\hat{V}_n^* - V^{\pi^*}) = \sqrt{n}(g_V(\hat{Q}_n) - g_V(Q)) \Rightarrow \mathcal{N}(0, \nabla g_V(Q)\Sigma(\nabla g_V(Q))^T) \text{ as } n \to \infty.$$

Note that $\nabla g_V(Q)$ is a $m_s \times N$ matrix with $\nabla g_V(Q)(i, (j-1)m_a + k) = \mathbb{1}(i = j, k = a^*(i))$. By rearranging the index such that $\tilde{P}^{\pi^*} = \left(\begin{array}{c|c} P^{\pi^*} & \mathbf{0} \\ \hline * & \mathbf{0} \end{array}\right)$ (where "$*$" denotes a placeholder of some quantities), we have $\nabla g_V(Q) = [I, 0]$, $D_R = Diag(D_R^{\pi^*}, *)$, $W = Diag(W^{\pi^*}, *)$ and $D_Q = Diag(D_V^{\pi^*}, *)$. We also note that

$$\begin{aligned} \nabla g_V(Q)(I - \gamma\tilde{P}^{\pi^*})^{-1} &= [I, 0]\sum_{i=0}^{\infty}\gamma^i(\tilde{P}^{\pi^*})^i \\ &= [I, 0]\sum_{i=0}^{\infty}\gamma^i\left(\begin{array}{c|c}(P^{\pi^*})^i & \mathbf{0} \\ \hline * & \mathbf{0}\end{array}\right) \\ &= \sum_{i=0}^{\infty}\gamma^i\left[(P^{\pi^*})^i, \mathbf{0}\right] = \left[(I - \gamma P^{\pi^*})^{-1}, \mathbf{0}\right]. \end{aligned}$$

Thus,

$$\begin{aligned} \nabla g_V(Q)\Sigma(\nabla g_V(Q))^T &= \left[(I - \gamma P^{\pi^*})^{-1}, \mathbf{0}\right]W^{-1}[D_R + D_Q]\left[(I - \gamma P^{\pi^*})^{-1}, \mathbf{0}\right]^T \\ &= (I - \gamma P^{\pi^*})^{-1}(W^{\pi^*})^{-1}[D_R^{\pi^*} + D_V^{\pi^*}]((I - \gamma P^{\pi^*})^{-1})^T. \end{aligned}$$

Lastly, the asymptotic normality of $\hat{\chi}_n^{\pi^*}$ follows from the continuous mapping theorem. $\square$

We next establish the asymptotic normality for the estimated value function under a given policy $\tilde{\pi}$. In this case, the value function $V^{\tilde{\pi}}$ satisfies a Bellman equation

$$V^{\tilde{\pi}}(s) = \sum_a \mu_R(s, a)\tilde{\pi}(a|s) + \gamma\sum_a\tilde{\pi}(a|s)\sum_{s'}P(s'|s, a)V^{\tilde{\pi}}(s').$$

Denote the the estimator of $V^{\tilde{\pi}}$ by $\hat{V}_n^{\tilde{\pi}}$. In particular, $\hat{V}_n^{\tilde{\pi}}$ is the fixed point of the corresponding empirical Bellman equation that replaces $(\mu_R, P)$ by $(\hat{\mu}_{R,n}, \hat{P}_n)$. We have:

**Corollary 2.** *Under Assumptions 1 and 3,*

$$\sqrt{n}(\hat{V}_n^{\tilde{\pi}} - V^{\tilde{\pi}}) \Rightarrow \mathcal{N}(0, \Sigma_V^{\tilde{\pi}})$$

*where*

$$\Sigma_V^{\tilde{\pi}} = X'W'X'^T,$$

$X' = (I - \gamma P^{\tilde{\pi}})^{-1}$, $P^{\tilde{\pi}}$ *is an $m_s \times m_s$ transition matrix with $P^{\tilde{\pi}}(i,j) = \sum_a P(j|s=i,a)\tilde{\pi}(a|s=i)$,*
$W'$ *is an $m_s \times m_s$ diagonal matrix with*

$$W'(i,i) = \sum_j \frac{\tilde{\pi}(j|i)^2}{w(i,j)}[(\gamma V^{\tilde{\pi}})^T \Sigma_{P_{i,j}}(\gamma V^{\tilde{\pi}}) + \sigma_R^2(i,j)]$$

*Proof of Corollary 2.* Similar to the proof of Theorem 1, define $F^{\tilde{\pi}}$ as a mapping from $\mathbb{R}^{m_s} \times \mathbb{R}^N \times \mathbb{R}^{Nm_s} \to \mathbb{R}^{m_s}$:

$$F^{\tilde{\pi}}(V', r', P')(s) = V'(s) - \sum_a r'(s,a)\tilde{\pi}(a|s) - \gamma \sum_a \tilde{\pi}(a|s) \sum_{s'} P'(s'|s,a)V'(s').$$

Note that $F^{\tilde{\pi}}(V^{\tilde{\pi}}, \mu_R, P) = 0$, $F^{\tilde{\pi}}$ is continuously differentiable and $I - \gamma P^{\tilde{\pi}}$ is invertible. We can thus apply the implicit function theorem. In particular, there exists an open set $U^{\tilde{\pi}}$ around $\mu_R \times P \in \mathbb{R}^N \times \mathbb{R}^{Nm_s}$, and a unique continuously differentiable function $\phi^{\tilde{\pi}} \colon U^{\tilde{\pi}} \to \mathbb{R}^N$, such that

$$\phi^{\tilde{\pi}}(\mu_R, P) = V^{\tilde{\pi}}$$

$$F^{\tilde{\pi}}(\phi^{\tilde{\pi}}(r', P'), r', P') = 0$$

for any $r' \times P' \in U^{\tilde{\pi}}$. For the partial derivatives of $\phi^{\tilde{\pi}}$, we have

$$\frac{\partial \phi^{\tilde{\pi}}}{\partial(r', P')}\Big|_{r'=\mu_R, P'=P} = -\left[\frac{\partial F^{\tilde{\pi}}}{\partial V'}\right]^{-1}\left[\frac{\partial F^{\tilde{\pi}}}{\partial r'}, \frac{\partial F^{\tilde{\pi}}}{\partial P'}\right]\Big|_{V'=V^{\tilde{\pi}}, r'=\mu_R, P'=P}$$

where

$$\frac{\partial F^{\tilde{\pi}}}{\partial V'}\Big|_{V'=V^{\tilde{\pi}}, r'=\mu_R, P'=P} = I - \gamma P^{\tilde{\pi}},$$

$$G^{\tilde{\pi}} := \frac{\partial F^{\tilde{\pi}}}{\partial r'}\Big|_{V'=V^{\tilde{\pi}}, r'=\mu_R, P'=P} = \begin{pmatrix} \tilde{\pi}(\cdot|1)^T & & & & \\ & \ddots & & & \\ & & \tilde{\pi}(\cdot|i)^T & & \\ & & & \ddots & \\ & & & & \tilde{\pi}(\cdot|m_s)^T \end{pmatrix},$$

and

$$H_V^{\tilde{\pi}} := \frac{\partial F^{\tilde{\pi}}}{\partial P'}\Big|_{V'=V^{\tilde{\pi}}, r'=\mu_R, P'=P} = \begin{pmatrix} (q_1^{\tilde{\pi}})^T & & & & \\ & \ddots & & & \\ & & (q_i^{\tilde{\pi}})^T & & \\ & & & \ddots & \\ & & & & (q_{m_s}^{\tilde{\pi}})^T \end{pmatrix}$$

where $(q_i^{\tilde{\pi}})^T = \gamma[\tilde{\pi}(1|i)(V^{\tilde{\pi}})^T, \dots \tilde{\pi}(j|i)(V^{\tilde{\pi}})^T, \dots \tilde{\pi}(m_a|i)(V^{\tilde{\pi}})^T]$, which is an $N$-dimensional row vector.

Applying the delta method, we have

$$
\begin{aligned}
\sqrt{n}(\hat{V}_n^{\tilde{\pi}} - V^{\tilde{\pi}}) &= \sqrt{n}(\phi^{\tilde{\pi}}(\hat{\mu}_{R,n}, \hat{P}_n) - \phi^{\tilde{\pi}}(\mu_R, P)) \\
&\Rightarrow \mathcal{N}(0, \nabla\phi^{\tilde{\pi}}(\mu_R, P)\Sigma_{R,P}\nabla\phi^{\tilde{\pi}}(\mu_R, P)^T) \text{ as } n \to \infty,
\end{aligned}
$$

where

$$
\begin{aligned}
& \nabla\phi^{\tilde{\pi}}(\mu_R, P)\Sigma_{R,P}\nabla\phi^{\tilde{\pi}}(\mu_R, P)^T \\
&= (I - \gamma P^{\tilde{\pi}})^{-1}[G^{\tilde{\pi}}, H_V^{\tilde{\pi}}] \left( \begin{array}{c|c} W^{-1}D_R & \mathbf{0} \\ \hline \mathbf{0} & D_P \end{array} \right) [G^{\tilde{\pi}}, H_V^{\tilde{\pi}}]^T ((I - \gamma P^{\tilde{\pi}})^{-1})^T \\
&= (I - \gamma P^{\tilde{\pi}})^{-1}(G^{\tilde{\pi}}W^{-1}D_R(G^{\tilde{\pi}})^T) + H_V^{\tilde{\pi}}D_P(H_V^{\tilde{\pi}})^T)((I - \gamma P^{\tilde{\pi}})^{-1})^T
\end{aligned}
$$

and the conclusion follows. □

*Proof of Theorem 2.* Write the MDP problem in its LP representation

$$
\begin{aligned}
&\max && \sum_s \rho(s)V(s) \\
&\text{subject to} && V(s) \geq r(s,a) + \gamma \sum_{s' \in S} P(s'|s,a)V(s'), \ \forall s,a
\end{aligned}
$$

with the dual problem

$$
\begin{aligned}
&\max && \sum_{s,a} \mu_R(s,a)x_{s,a} \\
&\text{subject to} && \sum_a x_{s,a} - \gamma \sum_{s',a} P(s|s',a)x_{s',a} = \rho(s), \ \forall s \\
& && x_{s,a} \geq 0, \ \forall s,a
\end{aligned}
$$

The decision variables in the dual problem, $x_{s,a}$'s, in particular represent the occupancy measures of the MDP. If the MDP has non-unique optimal policies, the dual problem also has non-unique optimal solutions, which implies a degeneration of the primal problem. Degeneration here means that some constraints are redundant at the primal optimal solution (i.e., the corner-point solution is at the intersection of more than $m_s$ hyperplanes). Since the rows of the primal LP are linearly independent, we know that in this case, there are multiple ($K' > 1$) choices for the set of basic variables $(v_k^B)_{1 \leq k \leq K'}$ at the optimal solution. When the coefficients in the intersecting hyperlanes perturb slightly along a given direction, the objective value will change by a perturbation of the objective coefficients along a chosen set of basic variables $v_k^B$. In other words, we can partition the set of directions $U$ into subsets $\{U_k\}_{1 \leq k \leq K'}$ such that, if the direction of perturbation of $(P, \mu_R)$, say $u$, lies in $U_k$, then the LP optimal value perturbs by fixing the basic variables as $v_k^B$. Denote $G_k^\rho$ as the gradient vector corresponding to this direction. If some of the $G_k^\rho$'s are equal, we merge the corresponding $U_k$'s into one partition set. Thus, we have $K^\rho \geq 1$ distinct $G_k^\rho$'s and a partition of $U = \cup_{1 \leq k \leq K^\rho} U_k^\rho$, where $U_k^\rho = \{u : D_u^\rho(P, \mu_R) = G_k^\rho\}$, and $D_u^\rho(P, \mu_R)$ denotes the gradient of $\sum_s \rho(s)V(s)$ with respect to $(P, \mu_R)$ along the direction $u$.

Note that the argument so far focuses on the LP with objective value $\sum_s \rho(s)V(s)$. However, we can repeat the same argument for each $V(s)$ by setting $\rho(s) = e_s$. For any $u \in U$ and $s \in \mathcal{S}$, denote the directional gradient of $V(s)$ with respect to $P, \mu_R$ by $D_u V(P, \mu_R)(s)$, thus we can define the directional Jacobian of $V$ with respect to $P, \mu_R$ as $D_u V(P, \mu_R) := [D_u V(P, \mu_R)(1), \ldots, D_u V(P, \mu_R)(s), \ldots, D_u V(P, \mu_R)(m_s)]^T$, which leads to $K \geq 1$ (potentially larger than $K^\rho$) distinct $G_k$'s, where $G_k \in \mathbb{R}^{m_s \times (Nm_s+N)}$ and $U = \cup_{1 \leq k \leq K} U_k$ is partitioned with $U_k = \{u : D_u V(P, \mu_R) = G_k\}$. Define $\hat{u}_n = (\hat{P}_n - P, \hat{\mu}_{R,n} - \mu_R)/\sqrt{||\hat{P}_n - P||^2 + ||\hat{\mu}_{R,n} - \mu_R||^2}$. We have

$$
\hat{V}_n^* - V^* = \sum_{k=1}^K G_k \mathbb{1}\left(\hat{u}_n \in U_k\right)(\hat{P}_n - P, \hat{\mu}_{R,n} - \mu_R) + o_P(||(\hat{P}_n - P, \hat{\mu}_{R,n} - \mu_R)||)
$$

Multiply $\sqrt{n}$ on both sides and notice that

$$\hat{u}_n = (\sqrt{n}(\hat{P}_n - P), \sqrt{n}(\hat{\mu}_{R,n} - \mu_R))/\sqrt{||\sqrt{n}(\hat{P}_n - P)||^2 + ||\sqrt{n}(\hat{\mu}_{R,n} - \mu_R)||^2}$$

is a continuous mapping of $(\sqrt{n}(\hat{P}_n - P), \sqrt{n}(\hat{\mu}_{R,n} - \mu_R))$. By taking $n \to \infty$, we get the result by the continuous mapping theorem.

$\square$

*Proof of Theorem 3.* We use the LP representation of the constrained MDP. Define $x_{s,a}$ as the occupancy measure

$$x_{s,a} = \sum_{t=0}^{\infty} \gamma^t P_\rho(S_t = s)$$

where $P_\rho$ denotes the distribution of $S_t$'s with initial distribution $\rho$. Then, $x_{s,a}$ satisfies the LP

$$
\begin{aligned}
\text{max} \quad & \sum_{s,a} \mu_R(s,a) x_{s,a} \\
\text{subject to} \quad & \sum_{s,a} \mu_C(s,a) x_{s,a} \leq \eta \\
& \sum_a x_{s,a} - \gamma \sum_{s',a} P(s|s',a) x_{s',a} = \rho(s), \ \forall s \\
& x_{s,a} \geq 0, \ \forall s, a
\end{aligned}
\tag{11}
$$

(This is the dual formulation in the proof of Theorem 2 with an extra constraint.) The objective and the first constraint correspond to the objective and constraint in the constrained MDP formulation. The second constraint can be deduced by a one-step analysis on the definition of occupancy measure. Once (11) is solved to obtain an optimal solution $(x^*_{s,a})_{s,a}$, it can be translated into an optimal policy

$$\pi^*(a|s) = \frac{x^*_{s,a}}{\sum_a x^*_{s,a}}$$

Note that the number of structural constraints in (11) is $m_s + 1$, and a corner-point optimal solution has $m_s + 1$ basic variables. Moreover, by our assumptions, the optimal solution is unique and the LP is non-degenerate, so that perturbing the parameters $\mu_R(s,a), \mu_C(s,a), P(s|s',a)$ does not immediately imply an overshoot to negativity for the reduced costs of the non-basic variables. Now consider two cases depending on whether the first constraint is non-binding or binding. The first case corresponds to a deterministic optimal policy, i.e., for any $s$, $x_{s,a} > 0$ for only one $a$. In this case a small perturbation of the parameters still retains the same basic and non-basic variables, and the derived perturbed policy still retains the non-binding first constraint. In this case, the analysis reduces back to Corollary 2.

In the second case, $x_{s,a} > 0$ for only one $a$, for all $s$ except one state $s_r$, where we can have $x_{s_r, a^*_1(s_r)} > 0$ and $x_{s_r, a^*_2(s_r)} > 0$ for two distinct actions $a^*_1(s_r), a^*_2(s_r)$. Again, perturbing the parameters retains these basic and non-basic variables. In particular, the first constraint remains binding in the perturbation, so that the perturbed optimal policy $\pi^*$ is still split at the same state and satisfies $\sum_s \rho(s) L^{\pi^*}(s) = \eta$. Now denote the mixing parameter by $\alpha^* := \pi^*(a^*_1(s_r)|s_r)$, and so $\pi^*(a^*_2(s_r)|s_r) = 1 - \alpha^*$. By applying the implicit function theorem to the Bellman equation, there exists a continuously differentiable function $\phi_L$ such that $L^{\pi^*}(s) = \phi_L(\mu_C, P, \alpha^*)$. By applying the implicit function theorem again to the equation $\sum_s \rho(s)\phi_L(\mu_C, P, \alpha^*) = \eta$, we know $\alpha^*$ is a continuously differentiable function of $\mu_C$ and $P$. Thus $V^*$ can be viewed as a function of $\mu_R$, $P$ and $\alpha^*$, the latter in turn depending on $\mu_C$ and $P$. It can also be viewed as a function of $\mu_R$, $\mu_C$ and $P$ directly. We use $\nabla_{\mu_R,\mu_C,P} V^*(\mu_R, \mu_C, P)$ to denote the Jacobian of $V^*$ with respect to $\mu_R$, $\mu_C$, $P$ when viewing $V^*$ as a function of these variables. We also use $\nabla_{\mu_R,\mu_C,P,\alpha} V^*(\mu_R, P, \alpha^*)$ to denote the Jacobian of $V^*$ with respect to $\mu_R, \mu_C, P, \alpha^*$, this time viewing $V^*$ as a function of $\mu_R, P, \alpha^*$.

We use similar notations throughout, and their meanings should be clear from the context. To facilitate derivations, we also distinguish the notation $\nabla_x f$, which denotes the multi-dimensional Jacobian matrix, from $\partial_x f$, which is used to denote the Jacobian when $x$ is a scalar (1-dimensional).

$$\nabla_{\mu_R,\mu_C,P} V^*(\mu_R, \mu_C, P) = \nabla_{\mu_R,\mu_C,P,\alpha^*} V^*(\mu_R, P, \alpha^*)[I, \nabla_{\mu_R,\mu_C,P}\alpha^*(\mu_C, P)^T]^T$$
$$= \nabla_{\mu_R,\mu_C,P} V^*(\mu_R, P, \alpha^*) + \partial_{\alpha^*} V^*(\mu_R, P, \alpha^*)\nabla_{\mu_R,\mu_C,P}\alpha^*(\mu_C, P) \quad (12)$$

Differentiating

$$\rho^T L^{\pi^*}(\mu_R, \mu_C, P) = \eta,$$

we have

$$\rho^T \nabla_{\mu_R,\mu_C,P} L^{\pi^*}(\mu_C, P, \alpha^*) + \rho^T \partial_{\alpha^*} L^{\pi^*}(\mu_C, P, \alpha^*)\nabla_{\mu_R,\mu_C,P}\alpha^*(\mu_C, P) = 0.$$

By rearranging the equation, we have

$$\nabla_{\mu_R,\mu_C,P}\alpha^*(\mu_C, P) = -\frac{1}{\rho^T \partial_{\alpha^*} L^{\pi^*}(\mu_C, P, \alpha^*)}\rho^T \nabla_{\mu_R,\mu_C,P} L^{\pi^*}(\mu_C, P, \alpha^*)$$

Substituting this into (12), we get

$$\nabla_{\mu_R,\mu_C,P} V^*(\mu_R, \mu_C, P) = \nabla_{\mu_R,\mu_C,P} V(\mu_R, P, \alpha^*)$$
$$- \frac{\rho^T \nabla_{\mu_R,\mu_C,P} L^{\pi^*}(\mu_C, P, \alpha^*)\partial_{\alpha^*} V^*(\mu_R, P, \alpha^*)}{\rho^T \partial_{\alpha^*} L^{\pi^*}(\mu_C, P, \alpha^*)}$$

Next, define an $m_s$-dimensional vector $r_C$ by $r_C(s) = \sum_{j=1}^{m_a} \mu_C(s,j)\pi^*(j|s)$. Then

$$\partial_{\alpha^*} L^{\pi^*}(\mu_C, P, \alpha^*) = \nabla_{r_C, P^{\pi^*}} L^{\pi^*}(r_C, P^{\pi^*})[(\partial_{\alpha^*} r_C(\alpha^*))^T, (\partial_{\alpha^*} P^{\pi^*}(\alpha^*))^T]^T$$

Note that $(I - \gamma P^{\pi^*})L^{\pi^*} = r_C$. By applying the implicit function theorem, we have

$$\partial_{\alpha^*} L^{\pi^*}(\mu_C, P, \alpha^*) = (I - \gamma P^{\pi^*})^{-1}\left[I, \begin{pmatrix} L^{\pi^*} & & & \\ & \ddots & & \\ & & L^{\pi^*} & \\ & & & \ddots & \\ & & & & L^{\pi^*} \end{pmatrix}\right] q_L^0$$

$$= (I - \gamma P^{\pi^*})^{-1} q_L,$$

where $q_L^0$ is a vector with $q_L^0(s_r) = \mu_R(s_r, a_1^*(s_r)) - \mu_R(s_r, a_2^*(s_r))$, $q_L^0(m_s + (s_r - 1)m_a + j) = P(j|s_r, a_1^*(s_r)) - P(j|s_r, a_2^*(s_r))$ for $1 \leq j \leq m_a$ and $q_L^0(i) = 0$ for any other index $i$.

Similarly, we can define $r_R$ by $r_R(s) = \sum_{j=1}^{m_a} \mu_R(s,j)\pi^*(j|s)$, and we have

$$\partial_{\alpha^*} V^*(\mu_R, P, \alpha^*) = (I - \gamma P^{\pi^*})^{-1} q_V$$

The derivation of $\nabla_{\mu_R,\mu_C,P} V^*(\mu_R, P, \alpha^*)$ and $\nabla_{\mu_R,\mu_C,P} L^{\pi^*}(\mu_C, P, \alpha^*)$ follows exactly the same line of analysis as how we derive $G^{\tilde{\pi}}$ and $H_V^{\tilde{\pi}}$ in the proof of Corollary 2. $\square$

*Proof of Theorem 4.* Denote

$$[\hat{\mu}_{R,n}, \hat{P}_n]_{S_0} = [\hat{\mu}_{R,n}((i-1)m_a + j), \hat{P}_n((i-1)N + (j-1)m_s + k)]_{i \in S_0, 1 \leq j \leq m_a, 1 \leq k \leq m_s},$$

and
$$[\mu_R, P]_{S_0} = [\mu_R((i-1)m_a + j), P((i-1)N + (j-1)m_s + k)]_{i \in S_0, 1 \leq j \leq m_a, 1 \leq k \leq m_s}.$$
By Assumption 6, we have

$$\sqrt{n}([\hat{\mu}_{R,n}, \hat{P}_n]_{S_0} - [\mu_R, P]_{S_0}) \Rightarrow \mathcal{N}(0, \Sigma_{R,P,S_0}) \text{ where } \Sigma_{R,P,S_0} = \begin{pmatrix} W_{S_0}^{-1} D_R^{S_0} & \mathbf{0} \\ \mathbf{0} & D_Q^{S_0} \end{pmatrix}.$$

Notice $M_g \circ M_I^{S_0} \circ \mathcal{T}_{\hat{\mu}_{R,n}, \hat{P}_n}$ only involves random variables $[\hat{\mu}_{R,n}, \hat{P}_n]_{S_0}$. Changing the distribution of $[\hat{\mu}_{R,n}, \hat{P}_n]_{S \setminus S_0}$ will not change the distribution of $\hat{Q}_n^M$. We can thus assign auxiliary random variables to $\hat{\mu}_{R,n}$ and $\hat{P}_n$ for all $i \notin S_0$, $1 \leq j \leq m_a$, $1 \leq k \leq m_s$. In particular, we define independent random variables for each $i \notin S_0$ by letting

$$\hat{\mu}_{R,n}((i-1)m_a + j) \stackrel{D}{=} \frac{1}{\sqrt{n}} \mathcal{N}(\mu_R((i-1)m_a + j), 1)$$

$$\hat{P}_n((i-1)N + (j-1)m_s + k) \stackrel{D}{=} \frac{1}{\sqrt{n}} \mathcal{N}(P((i-1)N + (j-1)m_s + k), 1).$$

By doing so, we extend the $m_{s_0} m_a$-dimensional random variable $[\hat{\mu}_{R,n}, \hat{P}_n]_{S_0}$ to an $m_s m_a$-dimensional random variable $[\hat{\mu}_{R,n}, \hat{P}_n]_S$ and

$$\sqrt{n}([\hat{\mu}_{R,n}, \hat{P}_n]_S - [\mu_R, P]_S) \Rightarrow \mathcal{N}(0, \Sigma_{R,P,S}) \text{ where } \Sigma_{R,P,S} = \begin{pmatrix} \Sigma_{R,P,S_0} & \mathbf{0} \\ \mathbf{0} & I \end{pmatrix}.$$

Similar to the proof of Theorem 1, define
$$F_M(Q', r', P') = Q' - M_g \circ M_I^{S_0} \circ \mathcal{T}_{r', P'}(Q').$$

By Assumption 4, $M_g$ is max-norm non-expansion. Then, $M_g \circ M_I^{S_0}$ is also max-norm non-expansion, implying that $\nabla(M_g \circ M_I^{S_0})$ has all its eigenvalues less than or equal to 1. Thus,

$$\frac{\partial F_M}{\partial Q'} = \nabla M(\mathcal{T}_{r', P'}(Q'))(I - \gamma \tilde{P}')$$

is invertible. By Assumption 4, we have $F_M(Q^M, \mu_R, P) = 0$. By Assumption 5, there exists a neighborhood $\Omega_M$ around $(Q^M, \mu_R, P)$, such that $F_M$ is continuously differentiable on $\Omega_M$. Then, applying the implicit function theorem, we have that there exists an open set $E_M \subset \Omega_M$ and a continuously differentiable mapping $\phi_M$ on $E_M$, such that $\phi_M(\mu_R, P) = Q^M$ and

$$\nabla \phi_M(\mu_R, P) = - \left[ \frac{\partial F_M}{\partial Q'} \right]^{-1} \left[ \frac{\partial F_M}{\partial r'}, \frac{\partial F_M}{\partial P'} \right] \Bigg|_{Q' = Q^M, r' = \mu_R, P' = P}.$$

Using the delta method, we have

$$\sqrt{n}(\hat{Q}_n^M - Q^M) \Rightarrow \mathcal{N}(0, \nabla \phi_M(\mu_R, P) \Sigma_{R,P,S} (\nabla \phi_M(\mu_R, P))^T) = \mathcal{N}(0, \Sigma_{S_0}^M).$$

$\square$

*Proof of lemma 1.* For any given policy $\pi$, by the balance equation for Markov Chains, its induced stationary distribution $w_\pi$ satisfies

$$\sum_{k,l} w_\pi((k-1)m_a + l) P(i|s = k, a = l) \pi(a = j|s = i) = w_\pi((i-1)m_a + j)$$

for any $i \in \mathcal{S}, j \in \mathcal{A}$. Summing up across $j$'s for each $i$, we have

$$
\sum_j w_\pi((i-1)m_a + j)
$$

$$
= \sum_j \sum_{k,l} w_\pi((k-1)m_a + l)P(i|s=k, a=l)\pi(a=j|s=i)
$$

$$
= \sum_{k,l} w_\pi((k-1)m_a + l)P(i|s=k, a=l)
$$

On the other hand, for any $w$ in $\mathcal{W}$, $\pi_w$ satisfies

$$
\sum_{k,l} w((k-1)m_a + l)P(i|s=k, a=l)\pi_w(a=j|s=i)
$$

$$
= \sum_{k,l} w((k-1)m_a + l)P(i|s=k, a=l)w((i-1)m_a + j)/\sum_u w((i-1)m_a + u)
$$

$$
= \sum_u w((i-1)m_a + u)w((i-1)m_a + j)/\sum_u w((i-1)m_a + u) = w((i-1)m_a + j)
$$

for all $i \in S$. Thus, $w$ is the stationary distribution of transition matrix $\tilde{P}^{\pi_w}$. $\qquad\square$

