# OpenReview forum: "Efficient Inference and Exploration for Reinforcement Learning"
_ICLR.cc/2020/Conference — Reject_

### Official Review · AnonReviewer3 · 2019-10-17
**Official Blind Review #3**

**Rating:** 1

**Review:**

Efficient Inference and Exploration for Reinforcement Learning
================================================================

This paper presents a pure-exploration algorithm for reinforcement learning.
The approach is based on an assymptotic analysis of the Q-values, and their convergence to central limit distribution.
Using this analysis, and under specific assumptions, the algorithm outperforms existing algorithms for exploration.


There are several things to like about this paper:
- Many existing analyses for efficient exploration are all based around more-of-the-same concentration bounds, which end up being quite messy and un-elegant. This approach based on central limit theorem appears to be more insightful and cleaner and I think there is something nice about that!
- The proposed algorithm is reasonable, for the setting in question, and the experimental results show that it outperforms benchmark exploration algorithms in this setting.
- The general structure of the paper, quality of writing and technical rigour appears to be of good standard... although I did not check all technical details carefully. [The paper is 24 pages long and we have many reviews]


However, there are also some significant places where this paper falls short:
- The problem setting that the authors consider is really not typical of the "exploration" problem in RL... I'm not talking about the fact that this is a "pure exploration" algorithm (that's fine), but instead that Assumption 3 is really not a good model for the types of problems that are "hard" for exploration in RL! For example, in the RiverSwim problem choosing an exploration policy = 0.8 right is essentially saying that you've already solved the hard part of the problem. Note - I am a little bit confused about the experiments in Tables 3 and 4, here it seems that you start with a pi(1|s)=0.6 which again feels like a cheat...
- Would it be possible to compare this algorithm in a more like-for-like standard RL setting, perhaps using the standardised bsuite https://github.com/deepmind/bsuite (the "deep sea" problems might be of particular interest here.)
- Alternatively, I can imagine a future version of the paper being more upfront about this deviation from the "standard" setting and highlighting that this is a special-type of result quite different from typical exploration in RL.
- I'm not sure that this sort of paper is well-served by a conference like ICLR... certainly there seems very little of "learning representation" in this discussion of Tabular RL. That would sort of be OK if the paper made nods to how these *insights* could carry over the deep learning or at least RL with (linear) function approximation... I don't see much of that.


Overall I do think this is an interesting paper, with a novel approach to pure exploration in tabular MDPs under specific assumptions.
However, I'm not sure that this paper is well-suited to ICLR and I have some concerns about whether it really does address the sort of "exploration" problem in RL that one might expect.

**Experience Assessment:**

I have published in this field for several years.

**Review Assessment: Checking Correctness Of Derivations And Theory:**

I assessed the sensibility of the derivations and theory.

**Review Assessment: Checking Correctness Of Experiments:**

I assessed the sensibility of the experiments.

**Review Assessment: Thoroughness In Paper Reading:**

I read the paper at least twice and used my best judgement in assessing the paper.

---

> ### Author Response · Authors · 2019-11-13
> **Response to review #3**
>
>
> Thank you for your comments and suggestions! We would like to reiterate, and as you have kindly pointed out, that our main contribution is to build a framework to tackle pure exploration in RL by developing a set of clean asymptotic results. To our knowledge, the considered pure exploration problem, even in the tabular MDP setting, has been under-studied yet arises in important applications. Our current work serves to build a foundational framework that potentially applies to more complex settings in the future, including some that you have suggested. In fact, in the Appendix, we discuss several possible extensions of our results, most noticeably on approximate value iteration in Appendix A2.
>
> Below are our point-by-point responses to your comments:
>
> 1. We would like to clarify that Assumption 3 is in some sense a minimal assumption for the ``correctness" of our framework (though it could be generalized to, say, non-irreducible MDP by suitable modification of our suggested strategy via properly randomized starting/restarting). On the other hand, we agree very much with you that this assumption does not capture the ``hardness" of the exploration problem. Rather, our approach is to propose a reasonable strategy backed by the implications and mathematical forms of our new limit theorems, and demonstrate its superiority through numerical experiments.
>     To clarify the specific numerical setup, $\pi(1|s) = 0.8$ is used in the first experiment using the RiverSwim example. The goal here is purely a sanity check of our statistical inference method, i.e., for any data collection mechanism, the 95% confidence interval constructed according to the formula is verified to contain the true value with a 95% chance when the sample size is large enough (asymptotic validity). We did not use this configuration as an exploration policy -- and we agree with the reviewer that doing so would defy the purpose of exploration.
>     In the second experiment (Table 3 and 4), $\pi(1|s)=0.6$ is used in the first stage to provide a warm start. The same first-stage data are used for all the methods listed in the table, either in estimating the parameters of the MDP (UCRL, Q-OCBA, $\epsilon$-greedy) or setting a good prior distribution for the parameters (PSRL); the pure random strategy RE(0.6) in the table keeps using $\pi(1|s)=0.6$ in the second stage. We observe in the experiments that using Q-OCBA outperforms $\pi(1|s)=0.6$ (thus improving a naive strategy) and other previous related methods in the literature (thus appearing competitive against existing alternatives; we also provide reasons for this in the numerical section).
>
> 2. Thank you for pointing out the other interesting test problems, which we would investigate in the future. We have chosen RiverSwim in this paper because it is the example used in the previous PSRL and UCRL papers. This problem has the specific feature that there are a lot of uncertainty about the transition probability, which elevates the need of exploration and thus we believe it is a good test problem for comparing exploration strategies.
>
> 3. Regarding your last two points, we would definitely clarify our setting and distinguish our framework/results from what was typically considered. In terms of the implications or generalizations to larger-scale problems, our framework that derives asymptotic results on estimation variance, and uses its mathematical form to formulate tractable optimization to devise good exploration strategies, is potentially applicable to these more sophisticated settings, and we do have them in our mind (e.g., Appendix A2 lists some preliminary results along this direction). However, a full investigation would require substantial additional analyses, and we would position the current paper as providing the groundwork for these important future generalizations.

---

> > ### Comment · AnonReviewer3 · 2019-11-14
> > **Thank you for your response**
> >
> > Overall I think that this distinction between the "exploration" studied here and the typical issues that arise in exploration is too much of a disconnection for me.
> >
> > I will have to have more of a think about this during the reviewer discussion, but at the moment I remain unconvinced to change my recommendation.

---

### Official Review · AnonReviewer1 · 2019-10-23
**Official Blind Review #1**

**Rating:** 3

**Review:**

This paper studies the asymptotic properties of action value function $Q$ and value function $V$. Specifically, the authors assume that we can collect $n$ data points $\{(s_t,a_t,r_t(s_t,a_t),s_{t+1})\}_{t=1}^{n}$. Based on the collected data, the authors calculated the sample mean of the unknown reward function $\widehat r_n$ and transition probability $\widehat{P}(\cdot|s,a)$. They further defined an estimator $\widehat Q_n$, which is the fixed point of the empirical Bellman operator $\widehat{\mathcal{T}}_n$ derived from $\widehat r_n$ and $\widehat P_n$. The authors proved under certain assumptions that $\widehat Q_n\rightarrow Q^*$ almost surely. Based on this argument, they also derived a similar convergence of value function $\widehat V_n$ in distribution. Confidence intervals can also be established based on these results.

The proof of convergence in distribution is established based on the following idea: $\widehat r_n$ and $\widehat P_n$ converge to $r$ and $P$ by central limit theorem. By examining the Jacobian matrix of $\widehat{\mathcal{T}}_n$ with respect to $\widehat Q$, it is found that the implicit function $\widehat{ \mathcal{T}}_n \widehat Q -\widehat Q=0$ can be solved in small neighborhoods of $\widehat r_n, \widehat P_n$ and $\widehat Q_n$ as
$$
\widehat{Q}_n = \phi(\widehat{r}_n, \widehat P_n) \qquad\text{(implicit function theorem)}
$$
Then by Slutsky theorem, $\widehat Q_n = \phi(\widehat r_n, \widehat P_n)$ also converges to $Q^*$ in distribution. The proof is straightforward and easy to follow. However, I have some concerns about the above analysis.

1. The data $\{(s_t,a_t,r_t(s_t,a_t),s_{t+1})\}_{t=1}^{n}$ are collected by executing a fixed policy for n steps, which means the data points are not i.i.d. Therefore, the convergence of  $\widehat r_n$ and $\widehat P_n$ by central limit theorem may not hold ground, which further leads the later proofs potentially problematic.

2. The $\phi(\cdot,\cdot)$ function in implicit function theorem only exists in neighborhoods of $\widehat r_n, \widehat P_n$ and $\widehat Q_n$. It is unclear from the current proof that $\widehat Q_n$ fall into the same neighborhood of $Q^*$. Therefore, it needs to be justified that the limit point is $Q^*$.

3. Even for the empirical estimator $\widehat Q_n$, it is still the fixed point of $\widehat{ \mathcal{T}}_n$, which is hard to compute or solve. Thus Algorithm 1 may be inefficient in practice.

Other comments:

The font and margin of this paper does not conform the format requirement of ICLR.
What is $\sigma_R(i,j)$ in Theorem 1 and Corollary 1? It seems to be undefined.

I found it hard to read when the authors vectorize all the matrices and tensors to $N$ dimension vectors and $N\times N$ matrices. In particular, the rearranged notations such $\mu((i-1)m_a+j)$, $\tilde P^{\pi}((i-1)m_a+j,(i’-1)m_a+j’)$ are much more complicated that $\mu(i,j)$ and $\tilde P^{\pi}(i’,j’|i,j)$. Similarly, the mapping $F(Q’,r’,P’)$ defined in the proof of Theorem 1 can be represented by the Bellman operator $\mathcal{T}$ defined on page 3.

====post rebuttal==
I read other reviewers' comments and the author's response. I do not think the authors have addressed all my concerns. I will keep my rating.

**Experience Assessment:**

I have published one or two papers in this area.

**Review Assessment: Checking Correctness Of Derivations And Theory:**

I assessed the sensibility of the derivations and theory.

**Review Assessment: Checking Correctness Of Experiments:**

I assessed the sensibility of the experiments.

**Review Assessment: Thoroughness In Paper Reading:**

I read the paper at least twice and used my best judgement in assessing the paper.

---

> ### Author Response · Authors · 2019-11-13
> **Response to review #1**
>
>
> 1. Our data is generated from a Markov Chain. There is a well-established central limit theorem for positive recurrent Markov chains.
>
>  2. $\hat{P}_n$ and $\hat{\mu}_{R,n}$ converges to $P$ and $\mu_R$ almost surely. By the implicit function theorem, $\phi$ is continuous, thus $\hat{Q}_n$ converges to $Q$ almost surely by the continuous mapping theorem. So, like Point 1 above, the concern of the referee has been rigorously handled in the paper.
>
> 3. Improving the computational efficiency in solving the Bellman equation, though interesting, is not a main focus of this paper, which is about quantifying the uncertainty due to data collection and designing exploration strategies to minimize these errors. We appreciate the reviewer's suggestion which is worth a good future work.
>
> 4. $\sigma_R(i,j)$ is defined in Assumption 1.
>
> 5. We need the vectorized notation since we apply the Delta method to $\hat{P}_n$ and $\hat{r}_{R,n}$, and the notation facilitates/simplifies subsequent discussions considerably. $F$ is also a necessary notation since we need to apply the implicit function theorem where the notion $\partial F$ is required.

---

### Official Review · AnonReviewer2 · 2019-10-24
**Official Blind Review #2**

**Rating:** 6

**Review:**

This paper studies the inference problem of reinforcement learning. With a given exploration policy that satisfies some strong property to collect n data, the paper studies the distribution of the estimated optimal value function and Q-function when n goes to infinity. Both unique and non-unique optimal policy cases are studied. The non-unique case has a very different behavior as it is no longer Gaussian. The paper then uses these estimations to design a method that better explores and proposes a method Q-OCBA. Experiments were performed to compare this method with previous algorithms, e.g., UCRL.

The inference results for the Q-function is definitely important and will be useful for the community. But the authors should not ignore all the finite-sample bound results in their related work section, especially for those who achieving minimax rates. For instance:
MG Azar, I Osband, R Munos, Minimax regret bounds for reinforcement learning, 2017
MG Azar, R Munos, H Kappen, Minimax PAC bounds on the sample complexity of reinforcement learning with a generative model, 2013
A Sidford, M Wang, X Wu, L Yang, Y Ye, Near-optimal time and sample complexities for solving Markov decision processes with a generative model, 2018
A Agarwal, S Kakade, L Yang, On the Optimality of Sparse Model-Based Planning for Markov Decision Processes

Please also see references therein. You can derive an inference result for the generative model as well.

More comments:
* The assumption on exploration policy is quite restrictive. It essentially says the bound does not hold if the MDP is not communicating. However, the Q-function is still well defined.
* The comparison to UCRL and other exploration-based method is not fair. In these results, the probability of picking "the" optimal policy is not important. Also, picking the correct action is not feasible if the problem is more complex. It is my understanding that your algorithm has a much larger regret due to that it requires to sample from every state.
* What about finite horizon MDP?



**Experience Assessment:**

I have published in this field for several years.

**Review Assessment: Checking Correctness Of Derivations And Theory:**

I assessed the sensibility of the derivations and theory.

**Review Assessment: Checking Correctness Of Experiments:**

I assessed the sensibility of the experiments.

**Review Assessment: Thoroughness In Paper Reading:**

I read the paper thoroughly.

---

> ### Author Response · Authors · 2019-11-13
> **Response to review #2**
>
>
> 1. Thank you for pointing out the line of studies on finite-sample bounds. We will incorporate this into the introduction section and properly relate it to our work.
>
> 2. Our results can be suitably modified to handle MDP where the underlying Markov chain is non-irreducible. In this case, the analysis would apply to the exploration policy designed for each commuting class. Alternatively, we can introduce randomized starting/restarting to create an irreducible chain.
>
> 3. Our understanding is that the reviewer thinks of the regret as the evaluation criterion, which has been used to design previous methods such as UCRL. In this paper we focus on a pure exploration problem instead of a regret analysis - There is no work to our knowledge that directly relates to this problem in the MDP setting (though there are analogs in the multi-arm bandit literature, namely best-arm identification). We are aware that methods like UCRL use a different criterion, but, besides very naive strategies, they are indeed the only strategies available for us to compare with. In fact, in our numerical section (last paragraph of page 8), we explained clearly that Q-OCBA outperforms PSRL and UCRL because the latter ones are designed for a different criterion, thus echoing the comment of the referee.
> 	One could certainly argue whether the probability of ``correct" selection (PCS) is a good criterion, or, more broadly, whether it makes sense to even study exploration without caring about the performance during the learning period. Though we acknowledge that in many interesting applications minimizing cumulative regret is a natural goal, there are also applications where efficient learning in an ``exploration phase"	without worrying about the cost of suboptimal decisions in the learning process is more appropriate. To our knowledge, ours is the first work considering this latter setting in RL. Our criterion of looking at PCS follows in some sense from the bandit literature. Again, one could consider replacing it by other criterion such as the expected value function gap, and also consider larger-scale problems where finding ``correct" action is infeasible. Our goal here is not to exhaust all these generalizations but to provide a new set of tools to start tackling this under-studied class of problems. Thank you for pointing out all these and we would love to investigate them in our future works.
>
>  4. We believe our framework can be extended to finite-horizon MDP, but it may warrant a separate work by itself. The idea of using the implicit function theorem and Delta method to characterize the asymptotic variance, and designing an exploration strategy by solving a tractable optimization to minimize this variance, still applies to the finite-horizon setting, but it would require a substantial new set of analyses.

---

### Decision · Program_Chairs · 2019-12-19

**Decision:**

Reject

**Comment:**

This paper examined a pure exploration method for efficient action value estimates in tabular reinforcement learning.  The paper is on the theoretical properties of value estimates in the large sample regime.  The method is shown to outperform baseline algorithms for this task in tabular reinforcement learning.

The reviewers were divided on the merits of this work.  The use of the central limit theorem was viewed as elegant, and the results were thought to be potentially useful.  However, the reviewers several limitations.  They found the assumption of a communicating MDP to be overly restrictive (reviewer 1).  The algorithm may be computationally inefficient (reviewer 2).  The nature of "exploration" in this work is not the conventional meaning in reinforcement learning (reviewer 3).

The paper is not yet ready for publication at ICLR.  The theoretical results do not clearly convey insights for reinforcement learning with function approximation, and the reviewers are also not in agreement that the current results are applicable to a general MDP setting.